# High-Q lasing via all-dielectric Bloch-surface-wave platform

Yang-Chun Lee [1], Ya-Lun Ho [1], Bo-Wei Lin [1], Mu-Hsin Chen [1], Di Xing[1], Hirofumi Daiguji [1] & Jean-Jacques Delaunay [1] ✉

Controlling the propagation and emission of light via Bloch surface waves (BSWs) has held promise in the field of on-chip nanophotonics. BSW-based optical devices are being widely investigated to develop on-chip integration systems. However, a coherent light source that is based on the stimulated emission of a BSW mode has yet to be developed. Here, we demonstrate lasers based on a guided BSW mode sustained by a gain-medium guiding structure microfabricated on the top of a BSW platform. A long-range propagation length of the BSW mode and a high-quality lasing emission of the BSW mode are achieved. The BSW lasers possess a lasing threshold of 6.7 μJ/mm² and a very narrow linewidth reaching a full width at half maximum as small as 0.019 nm. Moreover, the proposed lasing scheme exhibits high sensitivity to environmental changes suggesting the applicability of the proposed BSW lasers in ultra-sensitive devices.

Controlling the propagation and emission of light at the nanoscale has been considered one of the most challenging research topics, especially in the field of on-chip nanophotonics[1]. To achieve this purpose, the most popular strategy is to develop optical components based on surface electromagnetic waves, which exist at the interface between two different media having different dielectric constants[2]. Surface plasmon polaritons (SPPs) being one type of surface waves can be excited and propagate along a metal-dielectric interface. Due to SPPs properties including strong electromagnetic field enhancement, subwavelength confinement, and high surface sensitivity[2], manipulating light at a subwavelength scale through SPPs is made possible. However, the intrinsic Ohmic loss of metal reduces the propagation length of SPPs, rendering practical applications difficult. To address this critical issue, a hybridization of dielectric waveguiding light with plasmonics has been proposed using a configuration consisting of a dielectric waveguide separated from a metal surface by a thin insulating gap layer[3]. With this gap, a large portion of light energy will extend to the non-metallic regions so that such hybrid plasmonic modes can travel longer distances than those of the pure plasmonic modes, increasing the feasibility of SPPs applications. To date, hybrid plasmonic modes

have been widely investigated and applied in various fields such as on-chip subwavelength optics (e.g., waveguides[4] and nanolasers[5]) and sensors[6]. However, the propagation loss of the hybrid plasmonics is still high to be adapted in long-distance applications, which should be important for on-chip photonic integrated circuits (PICs). Moreover, the absorption loss also results in low-quality resonance in plasmonic nanolasers resulting in limited lasing performance[7].

Bloch surface waves (BSWs), another type of surface waves, are known to be confined at the interface between a periodically arranged dielectric multilayer and the surrounding medium[8], exhibiting strong field enhancement but the field localization is weaker than that of SPP[9]. It is noted that BSWs have several superior properties over SPPs, making them more favorable in various applications. For example, BSWs are sustained by a multilayer made of pure dielectric materials; they do not suffer from losses caused by metal absorption. Thus, BSWs can propagate very long distances (propagation length of centimeter range was obtained at visible wavelengths)[10] and achieve narrow mode resonances[9]. On the other hand, the dielectric multilayer can be prepared from different materials, allowing for BSWs generation covering a wide range of wavelengths from the deep ultraviolet (UV)[11] to mid-infrared (MIR) regions[12,13]. In contrast, although some metals such as

[1]Department of Mechanical Engineering, School of Engineering, The University of Tokyo, 7-3-1 Hongo, Bunkyo-ku, Tokyo 113-8656, Japan.
✉e-mail: jean@mech.t.u-tokyo.ac.jp

**Table 1 | Comparison table of photonic, hybrid plasmonic, and BSW-guided modes**

| | Photonic mode | Photonic mode (near cutoff) | Hybrid plasmonic mode | BSW mode |
|---|---|---|---|---|
| The cross-sectional schematic | |  |  |  |
| Electric field distribution |  |  |  |  |
| Thickness $d$ | 500 nm | 200 nm | 140 nm | 140 nm |
| Width $w$ | 1.50 μm | 1.50 μm | 1.50 μm | 1.50 μm |
| Confinement factor $\Gamma$ | 93% | 45.5% | 64% | 65% |
| $k_{effective}$ | ~0 | ~0 | $5.1 \times 10^{-3}$ | $7.8 \times 10^{-5}$ |

aluminum and platinum support SPPs in the UV to the visible region, their propagation losses are still too high to be used in practical devices[14].

According to the above-mentioned advantages of BSWs over SPPs, BSWs may have great potential for controlling the propagation and emission of light at the nanoscale; therefore, several studies about the investigation of their optical features and the development of optical devices based on BSW have emerged in recent years. For example, the propagation properties of BSWs within polymer stripes[15,16] or nanofibre[17] on the dielectric multilayer have been systemically investigated, and their guided BSW modes have been demonstrated. A lens-shaped ultrathin (100-nm-thick) polymer layer has been designed and fabricated to focus BSWs[18]. A circular grating structure has been proposed to demonstrate BSW coupling and focusing[19]. Two-dimensional (2D) disk resonator[20], as well as ring resonator[21–23] sustaining BSWs have been designed on the dielectric multilayer, and the optical resonance of BSWs has been realized experimentally[20,23]. In these studies, it is demonstrated that BSWs can be guided, deflected, diffracted, and even focused, through the appropriate design of in-plane structures on the dielectric multilayer (BSW platform), suggesting the feasibility of manipulating BSWs for various applications. To date, different types of BSW-based 2D optical devices have been developed, such as metallic waveguides[24] and couplers[25], Mach–Zehnder interferometers[26], beam splitters[27], light switching circuits[28], and optical logic gate[29], paving the way toward future on-chip integrated systems.

In past studies, the BSWs are usually excited externally by using planar prism-coupled configurations[12,15,18] or couplers[19,26–30]. To extend the applicability of BSWs in the field of on-chip nanophotonics, integrated light source are an indispensable component that should be developed and has been pursued for a long time. Several research groups have integrated different kinds of fluorescent materials (e.g., fluorescent dye doped in polymer[11], fluorescent proteins grafted onto polymeric waveguide[31], and quantum dots[32]) onto BSW platforms as light sources, and their emission properties, as well as the coupling into BSW propagating modes, have been studied. In addition, a resonant structure composed of organic dyes embedded in a circular Fabry-Pérot cavity on a dielectric multilayer has also been developed to emit BSW-coupled light having a spectral width below 1 nm with an enhanced emission rate through the Purcell effect[33]. However, these devices emit light by spontaneous emission of fluorescent materials and coherent light sources have not been demonstrated. Currently, an optical system consisting of halide perovskite nano- and microlasers and a BSW platform has been shown to exhibit highly directional and long propagation distances of the coupled light[34]. Although this work

demonstrated an integrated light source that emits coherent light, the laser light coupled to the BSW propagation mode is generated by a photonic lasing mode in a perovskite nano- or microwire cavity which remains difficult to integrate on a BSW platform due to its different mode nature and dimensions. Until now, a micro-fabricated coherent light source that is purely based on the stimulated emission of the BSW mode and compatible with on-chip integrated systems has yet to be developed.

Here, we demonstrate a top guiding structure-based BSW laser for the first time. The proposed BSW laser consists of only dielectric materials without any absorptive metals, thus avoiding Ohmic losses as in the plasmonic nanolasers, and at the same time realizing a large mode confinement factor. The laser cavity is fabricated on a dielectric multilayer platform that sustains the BSWs, and the lasing action is based on the stimulated emission of the BSW propagating mode. In this work, we first investigated three types of guided modes confined within polymer ridge waveguides on a quartz substrate, an Ag/SiO₂ substrate, and a dielectric multilayer (BSW platform), which correspond to the photonic mode, the hybrid plasmonic mode, and the BSW mode, respectively (Table 1). Although the photonic mode provides excellent optical confinement, the leakage of light into the substrate increases significantly when the thickness of the guiding structure is reduced to a dimension near the photonic cutoff thickness (about 200 nm). In contrast, both the hybrid plasmonic mode and BSW mode maintain good confinement when the thickness of the guiding structure is beyond the cutoff. However, since BSW modes do not suffer from Ohmic loss as the hybrid plasmonic mode, the simulated propagation loss of BSW mode is two orders of magnitude smaller than that of the hybrid plasmonic mode. Therefore, a high-quality lasing is achieved and should be highly promising for applications in long-range photonic circuits.

We realized this concept by preparing rhodamine 6 G (R6G)-doped SU-8 polymer ring cavities having a thickness below the cutoff thickness of the photonic mode onto the surface of a BSW platform. The proposed BSW lasers achieved a lasing threshold of 6.7 μJ/mm² as well as very narrow linewidth with a full width at half maximum (FWHM) as small as 0.019 nm, suggesting a very good quality resonance that is difficult to achieve using plasmonic nanolasers.

## Results

### Investigation and characterization of the BSW mode

Mode analysis (see details in the Methods section) is first employed to investigate and characterize the optical characteristics of the BSW mode (transverse electric-like guided mode) guided by an R6G-doped SU-8 polymer ($n = 1.6$) ridge waveguide having a thickness of 140 nm

on a dielectric multilayer. For comparison purposes, a photonic mode (transverse electric mode, guided by a 500- or 200 nm-thick polymer ridge waveguide on quartz) and a hybrid plasmonic mode (transverse magnetic mode, guided by a 140 nm-thick polymer ridge waveguide on Ag layer with a 10 nm SiO$_2$ interlayer) are also examined. Here, the wavelength is set to 580 nm, which is roughly centered at the lasing band observed in the R6G-doped SU-8 ring lasers (see Supplementary Fig. 1 in the Supplementary Information). The R6G is employed as the gain medium because of its high quantum yield for fluorescence[35]. The dielectric multilayer as a BSW platform comprises five pairs of alternating TiO$_2$ ($n = 2.31$) and SiO$_2$ ($n = 1.46$) layers having respective thicknesses of 71 and 154 nm. The details of the multilayer design are provided in Supplementary Fig. 2. Figure 1a–c show the calculated real part (Fig. 1a) and imaginary part (Fig. 1b) of the mode effective indices (expressed by $n_{effective} + ik_{effective}$, corresponding to the effective refractive index and the propagation loss) and the confinement factor (Fig. 1c) of the guided photonic, hybrid plasmonic, and BSW modes within the R6G-doped SU-8 ridge as a function of the thickness $d$ for a ridge width $w$ of 1.5 μm. The confinement factor is defined as the ratio between the electric energy in the gain medium (R6G-doped SU-8) and the total electric energy of the mode (see details in Methods). In Fig. 1a–c, the $k_{effective}$ of the photonic mode with a 500 nm-thick SU-8

ridge is almost zero (Fig. 1b), and its confinement factor is about 92.9% (Fig. 1c), meaning that the photonic mode is highly confined by the thick ridge waveguide as represented by the electric field distribution shown in Fig. 1d. When decreasing the ridge thickness, both the $n_{effective}$ and the $k_{effective}$ of the photonic mode exhibit a weak dependence (Fig. 1a, b). However, as displayed in Fig. 1c, the confinement factor of the photonic mode is dramatically decreased to less than 50% when the ridge thickness is reduced to below 200 nm. At around 180 nm, where the $n_{effective}$ approaches the refractive index of the quartz substrate, the light starts leaking into the quartz substrate so that the photonic mode cannot be supported by the ridge waveguide, as displayed in Fig. 1e. We attribute this behavior to the cutoff of the photonic mode, and this thickness ($d = 180$ nm) corresponds to the optical diffraction limit of the SU-8 ridge waveguide. In contrast to the photonic mode, the hybrid plasmonic mode and BSW mode both possess strong confinement factors at such a thin thickness of 180 nm, which are 66.6% and 73.0%, respectively (Fig. 1c), and the $n_{effective}$ of the respective guided modes supported by plasmonic and BSW waveguides are 1.67 and 1.38, respectively (Fig. 1a). When further decreasing the thickness of the ridge waveguide, the $n_{effective}$ and the confinement factor for the hybrid plasmonic mode and BSW mode continuously decreases, revealing a degradation of the confinement.

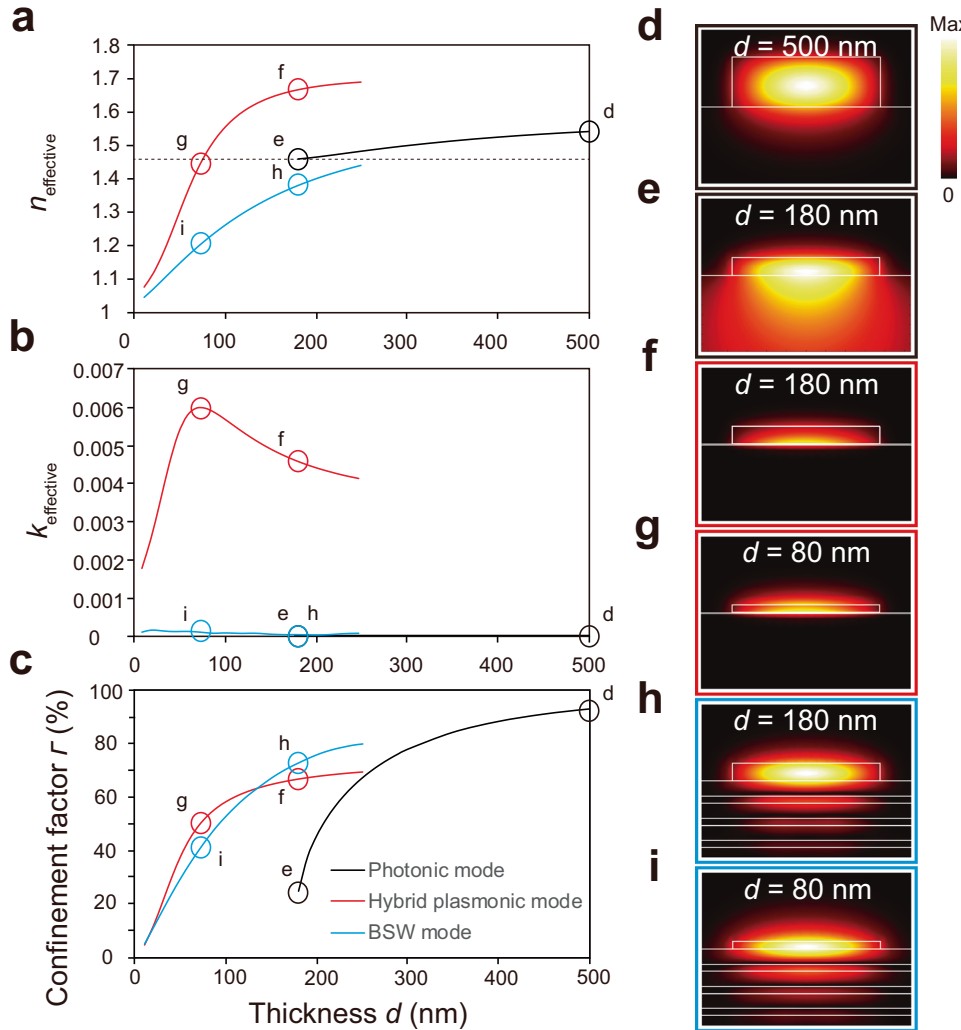

**Fig. 1 | Optical characteristics of photonic, hybrid plasmonic, and BSW modes as a function of the SU-8 polymer ridge thickness $d$ for a ridge width $w$ of 1.5 μm. a, b** The (**a**) real part and (**b**) imaginary part of the mode effective indices. **c** The confinement factor. **d–i** Calculated electric field norm distributions of the photonic mode for (**d**) $d = 500$ nm and (**e**) $d = 180$ nm, the hybrid plasmonic mode for (**f**) $d = 180$ nm and (**g**) $d = 80$ nm, and the BSW mode for (**h**) $d = 180$ nm and (**i**) $d = 80$ nm. The BSW platform is set to comprise five TiO$_2$/SiO$_2$ pairs having respective thicknesses of 71 nm/154 nm.

However, the two modes are still supported by the ridge waveguide even for thicknesses far beyond the cutoff thickness of the photonic mode (i.e., 80 nm), as displayed in Fig. 1f–i. Most importantly, the propagation loss ($k_{effective}$) of the BSW mode is two orders of magnitude less than that of the hybrid plasmonic mode within the entire thickness range of interest, as displayed in Fig. 1b. Furthermore, the effects of the ridge width on the optical characteristics of the photonic, hybrid plasmonic, and BSW modes are also reported in Supplementary Fig. 3. According to these results, we conclude that a BSW mode can provide good optical confinement at thicknesses below the cutoff thickness of the photonic mode similar to the hybrid plasmonic mode, while the BSW mode experiences significantly less propagation loss than the hybrid plasmonic mode. Therefore, the BSW mode is promising to be utilized for lasing provided that sufficient optical feedback is provided, and such BSW lasers will be more suitable for on-chip integrated systems due to their intrinsic low propagation losses.

## Experimental demonstration of the BSW lasers

The fabricated BSW lasers are shown in Fig. 2. The BSW laser devices consisted of ring laser cavities made of R6G-doped SU-8 polymer onto the surface of a BSW platform. The cavities were fabricated by top-down photolithography with a mask aligner (see details of the fabrication process in the Method section). As a comparison, we also fabricated the photonic lasers using the same R6G-doped SU-8 ring cavities on a quartz substrate. The ring cavities were designed and fabricated with different thicknesses ($d = 140$–500 nm), diameters ($D = 10$–100 μm), and ridge widths ($w = 1.5$–2.5 μm), as shown in Fig. 2b. Figure 2c shows the 5° tilted cross-sectional scanning electron microscopy (SEM) image of a fabricated R6G-doped SU-8 ring cavity

(see Supplementary Fig. 4). The thickness and width of the R6G-doped SU-8 ridge are characterized by atomic force microscopy (AFM), as displayed in Fig. 2d. Note that according to the SEM and AFM images, the ridge walls are not vertical (inset in Fig. 2c and Supplementary Fig. 5), and the effect of the oblique sidewall is discussed in Supplementary Fig. 6. It was also found that the actual thicknesses of the $TiO_2$ and $SiO_2$ layers are 82 and 170 nm, respectively, so both layers are slightly thicker than their designed values. The BSW mode can still be sustained by such a dielectric multilayer, and the effect of the thicknesses on the optical characteristics of the BSW mode will be discussed later.

The BSW ring lasers were optically pumped by a nanosecond laser at a wavelength of 532 nm at room temperature (see Method and Supplementary Fig. 7). Figure 3a shows the emission spectra of a BSW ring laser cavity having a diameter of 100 μm, a ridge thickness of 136 nm, and a ridge width of 1.5 μm under different pump energy densities. The output emission intensity as a function of pump energy density is shown in Fig. 3b and reveals the onset of the nonlinear lasing threshold ($P_{th}$) at 31.4 μJ/mm². The lasing of the BSW ring laser with a very large diameter ($D = 100$ μm) confirms the long-range propagation property of the guided BSW mode (propagation length longer than 100 μm), which is not possible for structures based on plasmonic waveguides. It is noted that the smallest FWHM of 0.019 nm is obtained from the BSW laser having a diameter of 50 μm, and the real FWHM could be considered smaller than this value due to the equipment limitation (Fig. 3c; also see Supplementary Fig. 8). This spectrograph-limited FWHM suggests that the BSW lasers exhibit excellent lasing performance due to the high-quality resonance of BSW-based modes achieving narrower linewidths than the reported

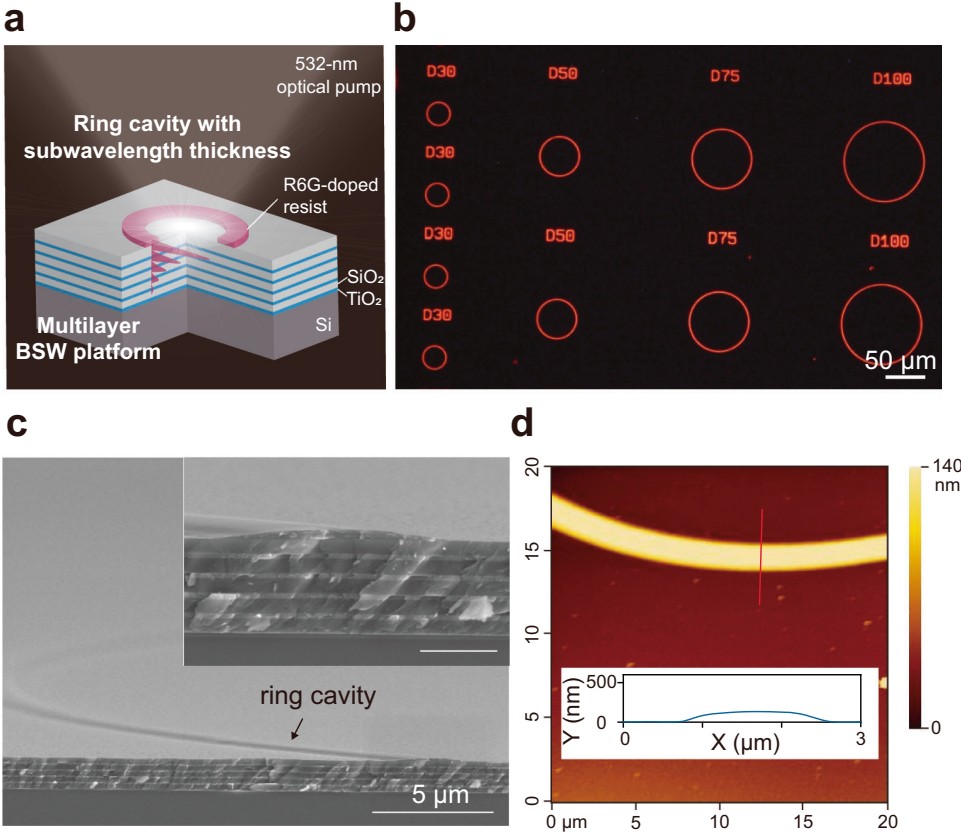

**Fig. 2 | The BSW laser. a** Schematic representation of the laser device consisting of an R6G-doped SU-8 140 nm-thick ring cavity onto a $TiO_2/SiO_2$ multilayer BSW platform. **b** μ-PL image of the ring laser cavities. **c** Tilted cross-sectional SEM image (tilt angle of 5°) of a ring cavity having a thickness of 140 nm and a diameter of 100 μm onto the surface of the BSW platform. Inset: magnified SEM image of the ring cavity with a scale bar of 1 μm. **d** AFM image of the R6G-doped SU-8 ridge of the ring cavity onto the surface of the BSW platform. Inset: the height profile along the red solid line.

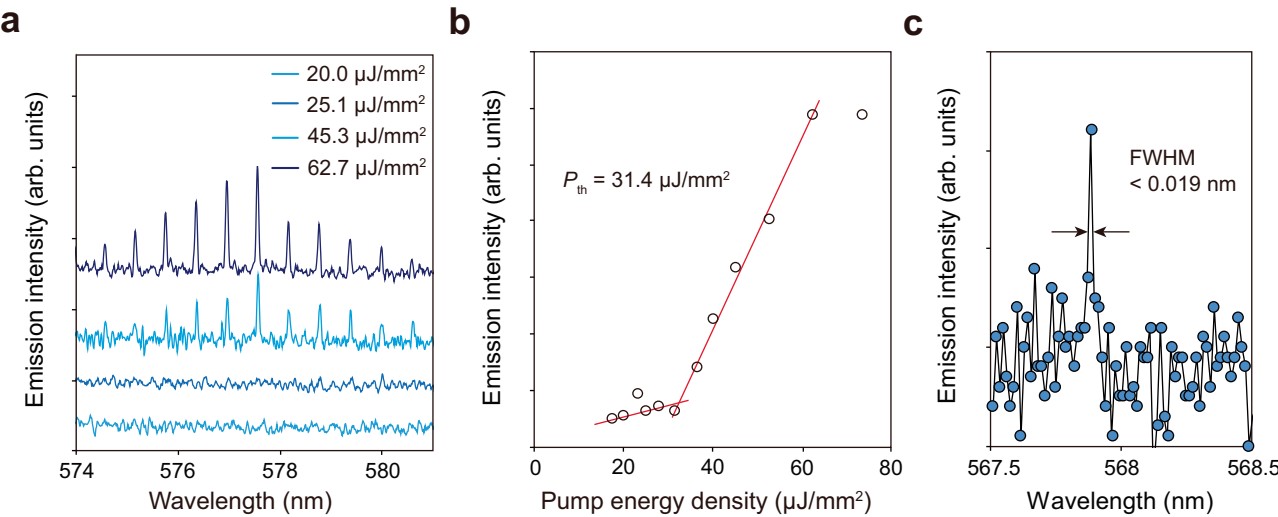

**Fig. 3 | Lasing performance of BSW lasers. a** Emission spectra under different pump energy densities and (**b**) output emission intensity as a function of pump energy density for a ring cavity having $d = 136$ nm, $w = 1.5$ μm, and $D = 100$ μm. **c** The magnified emission spectrum of a BSW laser having $d = 136$ nm, $w = 1.5$ μm, and $D = 50$ μm.

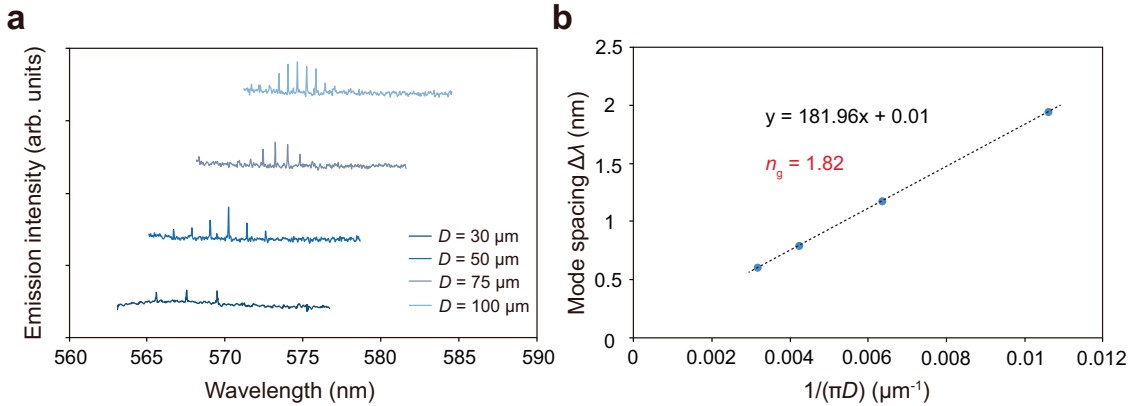

**Fig. 4 | Mode control of the BSW lasers. a** Emission spectra of BSW lasers with different diameters for a ring cavity with $d = 136$ nm and $w = 1.5$ μm. **b** Mode spacing analysis of the BSW lasers as a function of their cavity dimensions. The mode spacing is inversely proportional to the diameter of the ring cavity.

plasmonic micro/nanolasers (see Supplementary Table 1 in the Supplementary Information)[5,36–50].

The control of the lasing modes of BSW lasers is investigated in the following. Figure 4a displays the lasing spectrum of the BSW lasers having different diameters ranging from 30 to 100 μm. Note that the BSW ring cavities having diameters of 10, 15, and 20 μm did not lase. The central wavelength of the lasing spectrum was found to be dependent on the diameter of the ring cavities: the central wavelength was around 567 nm for a ring diameter of 30 μm and redshifted to around 575 nm for a larger diameter of 100 μm. This size dependence is attributed to two possible reasons. First, the lasing wavelength range can vary due to the change in the wavelength for the maximum out-coupled lasing light intensity from the ring cavity, which shifts toward longer wavelengths when the ring diameter is increased[51]. Second, the variation in the lasing wavelength range can be explained by the intrinsic self-absorption in the ring cavity[52]. In addition, the lasing mode spacings $\Delta\lambda$ also depended on the diameter of ring cavities, as observed in Fig. 4a. The mode spacing $\Delta\lambda$ versus the inverse of the cavity dimension $1/(\pi D)$ is plotted in Fig. 4b. A linear dependence between $\Delta\lambda$ and $1/(\pi D)$ is observed in accordance with a whispering-gallery-mode (WGM) resonance. The group index $n_g$ of the BSW lasers was estimated to be 1.82 from the theoretical

prediction $\Delta\lambda = \lambda^2/(\pi D n_g)$[53]. This value agrees well with the calculated $n_g$ obtained from the dispersion of the BSW mode and is significantly different from those of a photonic mode (see Supplementary Fig. 9). Thus, these experimental results on the group index confirm the BSW lasing behavior of the proposed BSW lasers. According to these results, the lasing wavelengths and the mode spacings of the BSW lasers can be controlled by modifying the size of the ring cavities.

To confirm lasing of the BSW ring cavity, corresponding photonic ring cavities having the same structural parameters ($D = 30$–$100$ μm, $w = 1.5$–$2.5$ μm) with the various thicknesses of $d = 136$, $205$, $240$, $290$, $310$, and $520$ nm were fabricated on a quartz substrate and characterized (see Supplementary Fig. 10). The lasing is not observed with the cavity thickness below 240 nm. In the 240 nm-thick ring cavities, only the 100 μm-diameter ring cavities with $w = 2$ and $2.5$ μm show lasing behavior with the thresholds of 96.1 and 98.3 μJ/mm², respectively, as seen in Fig. 5a. A further increase in the thickness results in lower lasing thresholds (see Supplementary Fig. 10). Accordingly, the lasing behavior of the ring cavities on the quartz substrate agrees with the mode analysis for a photonic mode (Fig. 1), showing less optical confinement and a mode cutoff when the thickness is reduced. Because the lasing emissions can still be found from the

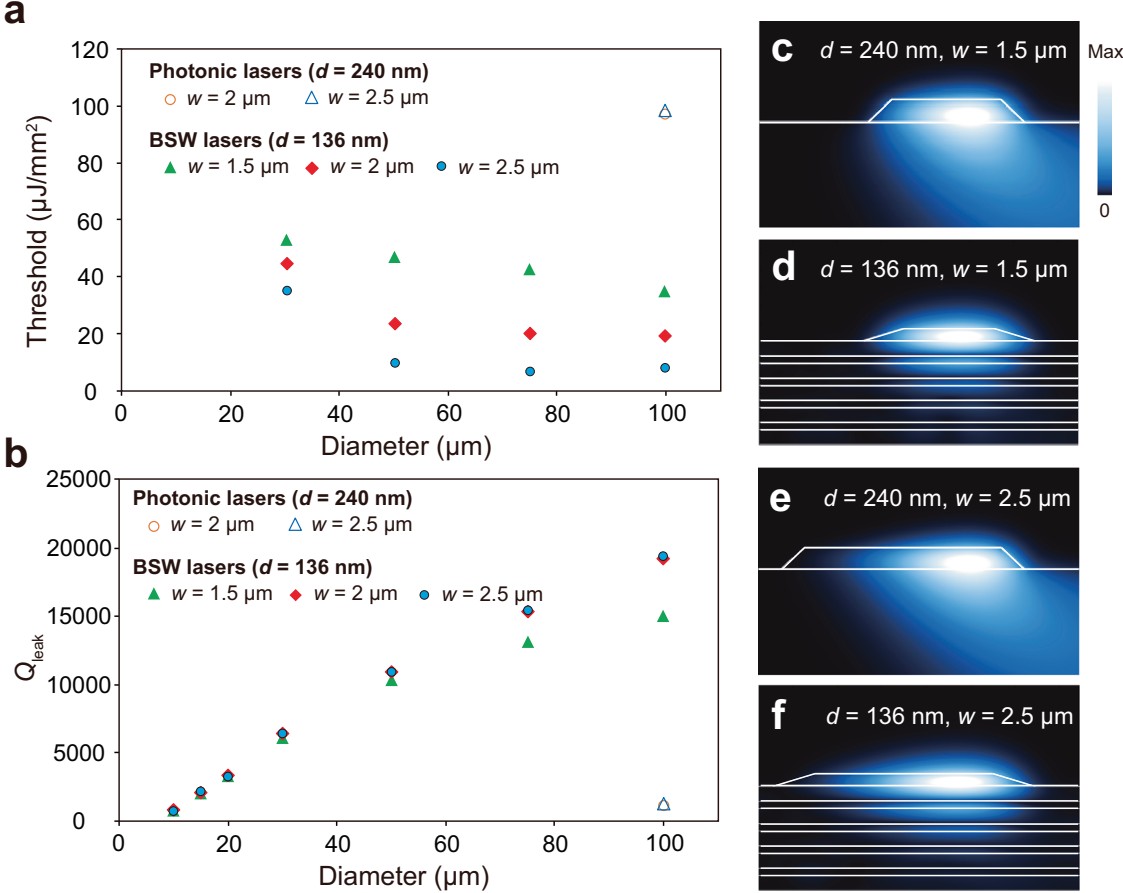

**Fig. 5 | Lasing threshold investigation of BSW lasers. a** The measured lasing threshold values of BSW lasers. The thresholds of two photonic lasers are also displayed. **b** The simulated leakage-related quality factor (defined as $Q_{leak}$) of the ring laser cavities. Note that the BSW ring cavities having diameters of 10, 15, and 20 μm did not lase. **c**–**f** The calculated electric field norm distributions for the photonic ring lasers with (**c**) $d = 240$ nm and $w = 1.5$ μm, and (**e**) $d = 240$ nm and $w = 2.5$ μm, and for BSW ring lasers with (**d**) $d = 136$ nm and $w = 1.5$ μm, and (**f**) $d = 136$ nm and $w = 2.5$ μm obtained by eigenfrequency analysis of an axisymmetric model. The diameters of the ring cavities are 100 μm. The BSW platform is set to comprise five TiO₂/SiO₂ pairs having respective actual thicknesses of 82 nm/170 nm as determined by SEM.

thin ring cavities with $d = 136$ nm on the BSW platform, the gain mechanism of the BSW lasers is indeed based on the stimulated emission of a BSW mode rather than a photonic mode. To confirm BSW lasing, discussions are provided about the higher modes in Supplementary Fig. 11 and the internal modes in Supplementary Fig. 12, and experimental evidence is also provided by leakage radiation microscopy in Supplementary Fig. 13.

**Investigation of the lasing thresholds of the BSW lasers**
Although the photonic-mode laser cavities with $d = 240$ nm, which can achieve lasing, possess similar optical confinements as BSW-mode laser cavities with $d = 136$ nm as displayed in Fig. 1c and Supplementary Fig. 3c, the BSW lasers show much lower lasing thresholds (~7–53 μJ/mm²) than that of the photonic lasers (~98 μJ/mm²) (Fig. 5a). To better understand the gain mechanism of the BSW lasers, we investigated the quality factors of the ring cavities. Particularly, the total quality factor $Q_{total}$ of an optical cavity is limited by several loss mechanisms and can then be expressed by[54]

$$\frac{1}{Q_{total}} = \frac{1}{Q_{abs}} + \frac{1}{Q_{leak}} + \frac{1}{Q_{scat}}, \tag{1}$$

where $Q_{abs}$ is the quality factor considering the absorption losses, $Q_{leak}$ the quality factor related to the lateral radiation leakage due to the effect of curvature and the leakage of light to the air or the substrate

(i.e., surrounding) during the light propagation, and $Q_{scat}$ the quality factor due to scattering losses. The quality factor refers to the energy loss of light propagation in an optical cavity and is therefore related to the threshold of a laser cavity[54].

Due to the transparent nature of the SU-8 polymer, the absorption loss ($1/Q_{abs}$) of the ring cavities should be low so that it can be neglected in the analysis of loss mechanisms[55]. On the other hand, the leakage-related quality factor $Q_{leak}$ can be estimated using the eigenfrequency analysis for an axisymmetric model (see simulations details in Methods). Here, the SU-8 polymer ridges in the simulation model are set to possess non-vertical sidewalls as observed from the SEM and AFM images (Fig. 2c, d) to take account of the effect of the oblique sidewalls. The $Q_{leak}$ is denoted by

$$Q_{leak} = \frac{\omega_{real}}{2\omega_{imag}}, \tag{2}$$

where $\omega_{real}$ and $\omega_{imag}$ are the real part and imaginary part of the calculated eigenfrequency, respectively. Figure 5b shows the simulated $Q_{leak}$ of the BSW ring cavities ($d = 136$ nm, $w = 1.5$, 2, and 2.5 μm) and photonic ring cavities ($d = 240$ nm, $w = 2$ and 2.5 μm) for different diameters. The $Q_{leak}$ values of the photonic ring cavities are relatively low ($Q_{leak}$ ~ 1200) compared to those of the BSW laser ring cavities ($Q_{leak}$ ~ 6000–20,000), which suggests that the photonic lasers ($d = 240$ nm) should exhibit higher lasing thresholds than that of the

BSW lasers ($d = 136$ nm) even though the photonic lasers possess similar confinement factors in the mode analysis. Note that although the $Q_{leak}$ values of the ring cavities with $D = 15$ and 20 μm ($Q_{leak} \sim 2000$ and 3300) are higher than those of the photonic ring cavities ($Q_{leak} \sim 1200$), lasing could not be observed thus suggesting the presence of other losses (e.g., scattering losses) limiting the lasing performance as discussed in the later section. Also, the $Q_{leak}$ values of the BSW ring cavities increased with increasing diameter, due to the reduced curvature effect when the light propagates along a curved ridge with smaller curvature for the WGM resonance. Figure 5c–f show the simulated electric field norm distributions for the WGM resonances of the photonic ring cavities ($d = 240$ nm, $D = 100$ μm) with $w = 1.5$ and 2.5 μm shown in Fig. 5c,e, respectively, and the BSW ring cavities ($d = 136$ nm, $D = 100$ μm) with $w = 1.5$ and 2.5 μm shown in Fig. 5d, f, respectively, using eigenfrequency analysis for an axisymmetric model. The WGM modes in the photonic ring cavities experience very high light leakage into the quartz substrate (Fig. 5c, e), resulting in large losses and low-quality factors. In contrast, the WGM modes in the BSW ring cavities are well supported by the ring cavity so that the radiation leakage remains low (Fig. 5d, f). In summary, the photonic lasers ($d = 240$ nm) possessed much higher lasing thresholds than the BSW ring lasers ($d = 136$ nm) due to high radiation leakage.

In Fig. 5a, the measured thresholds show a strong dependence on the ridge width, which is different from the trend observed in simulated $Q_{leak}$, as seen in Fig. 5b. To understand this difference, the losses of the BSW ring cavities are further investigated as follows.

According to the definition, $Q_{leak}$ is the quality factor related to the lateral radiation leakage due to the effect of curvature (denoted as $Q_{leak\text{-}curv}$) and the leakage of light to the surrounding (denoted as $Q_{leak\text{-}surr}$). To investigate the light leakage to the surrounding, $Q_{leak\text{-}surr}$ is evaluated using the propagation losses of the corresponding guided BSW mode in a straight ridge waveguide[21], and is defined and expressed via the mode effective indices ($n_{effective} + ik_{effective}$) by[56]

$$Q_{leak-surr} = \frac{n_{effective}}{2k_{effective}}. \tag{3}$$

The $Q_{leak\text{-}surr}$ of the BSW ring cavities are then calculated for the ridge with non-vertical sidewalls as a function of width (see Supplementary Fig. 14). Because the smaller ridge width provides less lateral optical confinement for the BSW mode (see Supplementary Fig. 15a–c), $Q_{leak\text{-}surr}$ decreases with decreasing ridge width. Note that the values of $Q_{leak\text{-}surr}$ (~10,000–40,000) are of the same order as the $Q_{leak}$ of the BSW ring cavities. The detailed investigation reveals that the light leakage to the surrounding is indeed a dominant loss (see Supplementary Figs. 16 and 17) so that their $Q_{leak}$ exhibit a weak ridge width dependence, which is different from the ridge width dependence on the lasing thresholds of the BSW lasers (Fig. 5b). We attribute this dependence to some losses (e.g., surface scattering loss) which exhibit strong ridge width dependence as discussed in the following.

In past studies, the $Q_{scat}$ is estimated to be greater than $10^6$ for the polymeric cavities fabricated by lithography techniques[56,57]. However, for the presented BSW mode which possesses weaker field localization, the scattering from the edge or surface roughness should be more significant. So, we suspect the ridge width dependence on the measured thresholds of BSW lasers to be due to surface scattering losses. Although the root-mean-square roughness of the fabricated R6G-doped SU-8 polymer ridge is likely independent of the ridge width (measured by AFM, see Supplementary Fig. 18), the electric field distributions of the guided BSW mode within the ridge become less confined with decreasing the ridge width, implying that more light can interact with the edge or surface roughness (as we are attempting to explain the lasing threshold, the effect of the roughness on the losses modeled by the mode analysis can be used to discuss the ridge width dependence, considering the pumping process which includes the

propagation of the BSW-guided mode before the establishment of the WGM resonance mode; see Supplementary Fig. 15d–f). Accordingly, we attribute the surface scattering losses of the BSW laser cavities to the possible reason for the strong ridge width dependence on the measured thresholds.

So far, we have investigated the loss mechanisms of the BSW lasers, which are found to strongly depend on the optical confinement of the BSW-guided mode. It is also known that losses can be decreased using a multilayer design with a red-shifted forbidden band[16], a strategy employed to obtain improved lasing performance (see Supplementary Figs. 19 and 20).

## Potential applications of the proposed BSW lasers

In the following, we discuss possible applications of the proposed BSW lasers by first showing the possibility for on-chip integration and then reporting a sensitivity investigation. The coupling between a BSW ring laser cavity and a waveguide, which is required for on-chip integration, is investigated by simulation in Supplementary Fig. 21.

Finally, we experimentally performed a sensitivity investigation to environmental changes of the proposed BSW lasers by monitoring the lasing wavelength in the presence of toluene vapor (experimental procedure described in the Supplementary Information). Before performing the sensitivity study, the stability of the BSW laser is inspected (regarding the photobleaching effect, see Supplementary Fig. 22). Figure 6a displays the emission spectra of a BSW ring laser collected at a time interval of 1 h and reveals very stable lasing spectra with no peak shift and

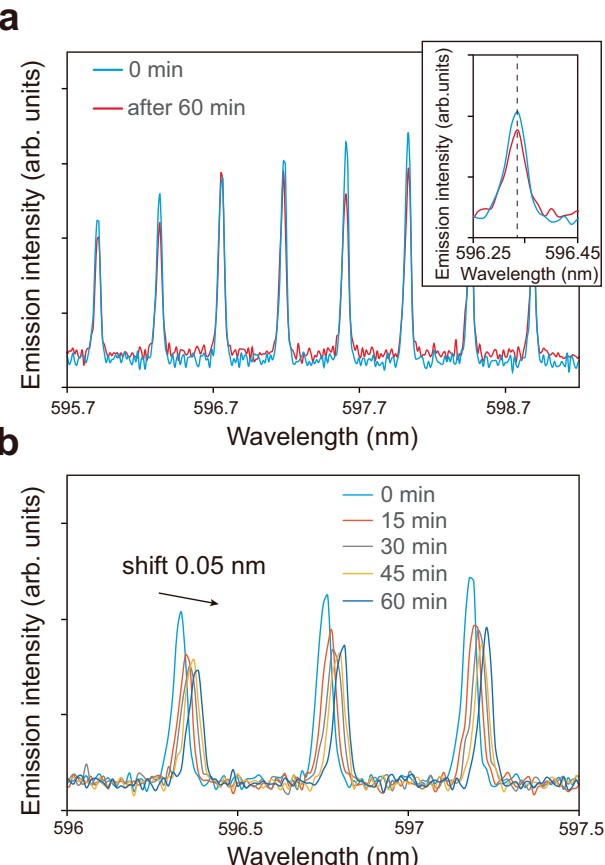

**Fig. 6 | Sensitivity investigation of BSW lasers. a** Stability over time of emission spectra of a BSW ring laser having a diameter of 100 μm: initial spectrum (0 min, blue) and spectrum after 1 h (60 min, red). Inset: The magnified graph of a lasing peak over time showing high stability. **b** Variation of the lasing peaks over time when the BSW ring laser is exposed to toluene vapor.

change in linewidth. In contrast, when the BSW laser was exposed to toluene vapor, the lasing peaks of emission spectra clearly shifted over time, as seen in Fig. 6b. The lasing peak shift is explained by the sorption of toluene vapor only or the sorption followed by swelling of the SU-8 resist due to toluene molecules diffusing into the SU-8 polymer[58] (regarding the clarification of the sensing mechanism, see Supplementary Figure 23). The sorption of toluene vapor onto the surface of SU-8 polymer increases the refractive index of the adjacent medium; the swelling of SU-8 polymer is known to increase the resist volume and refractive index, thus shifting the lasing peaks[59]. Due to the stability and narrow FWHM of the lasing peaks, the redshift of the lasing peaks can be precisely observed with an increment of ~0.012 nm. A total redshift of 0.05 nm is finally observed which can correspond to a sorbed toluene layer of 0.14 nm, an increase in the ridge size of 0.1 nm or a change in the refractive index of 0.00023 (see Supplementary Fig. 24), thus implying the detection of a very small perturbation of the BSW laser structure and suggesting possible applications of the BSW lasers as ultra-sensitive devices.

## Discussion

We have proposed and demonstrated a BSW laser whose lasing action is based on the stimulated emission of the BSW propagation mode. Mode analyses of the propagation properties of the photonic, hybrid plasmonic, and BSW-guided modes within a SU-8 ridge waveguide revealed that the BSW mode possessed similar optical confinement as the hybrid plasmonic mode for thicknesses of the ridge waveguide well below the cutoff thickness of the photonic mode. More importantly, the BSW mode exhibited lower propagation losses than that of the hybrid plasmonic mode, thus making it possible for the BSW mode to lase as long as sufficient optical feedback is provided. The BSW laser devices were realized by fabricating R6G-doped SU-8 microring cavities onto the surface of a BSW platform. Optical pumping using a nanosecond pulsed laser of the BSW lasers revealed lasing thresholds (6.7 μJ/mm$^2$) and narrow linewidths (0.019 nm corresponding to the spectrometer spectral resolution). These results confirmed that the BSW lasers exhibited very good quality resonances that cannot be achieved with plasmonic-based nanolasers. The variation of the lasing threshold with the ring diameter and ridge width indicated that the energy losses of the BSW modes propagating in the ring cavity strongly depend on their structural parameters, providing an insight for optimization of the lasing performances of the BSW lasers. Minute changes in the surrounding of the BSW lasers were found to induce lasing peak shifts as small as 0.01 nm, indicating the possibility to develop a high-sensitivity detection scheme based on the BSW lasers. From the above results, the proposed BSW lasers are thought to hold great potential for applications in ultracompact on-chip integrations and high-throughput sensors due to their high-quality resonances and low propagation losses.

## Methods

### Sample preparation

The fabrication of the ring cavities included two parts: the preparation of the photomask and the top-down photolithography with a mask aligner. To prepare the photomask, a chromium (Cr) layer having a thickness of 100 nm was deposited on a quartz substrate using a magnetron sputtering system. After the deposition of the Cr layer, a laser lithography process was conducted to define the pattern of the ring cavities. First, a hexamethyldisilazane (HMDS) layer was spin-coated onto the Cr layer at 4000 rpm for 30 s and then baked at 110 °C for 1 min. Next, a 1000 nm-thick positive photoresist (7790 G, JSR Corporation) was spin-coated onto the HMDS layer at 6000 rpm for 30 s, followed by baking at 110 °C for 1.5 min. Then, the patterns of the ring cavities were written with a laser lithography system (DWL 66+ laser writer, Heidelberg instruments), and developed by an NMD-3

solution for 60 s, followed by rinsing with water for 10 s twice and drying with airflow. After the laser lithography process, the patterned photoresist was used as a mask for the wet etching of the Cr layer. The etching process was conducted by soaking the substrate into a Cr etchant for 1 min, followed by rinsing with water for 10 s twice and drying in an airflow.

The top-down photolithography was conducted by the following steps. First, the Rhodamine 6 G laser dye (R6G, dye content ~95%, Sigma-Aldrich) was dissolved in ethanol to form a 10 mM solution. Then, the dye-containing solution was mixed with SU-8 (SU-8 2000.5, solid content 14.5%, KAYAKU Advanced Materials) and cyclopentanone (≥99%, Sigma-Aldrich) solution at a ratio of 10% (R6G + ethanol) to 90% (SU-8 + cyclopentanone) in the ultrasonic bath for 60 min. Here, cyclopentanone was used as a thinner for SU-8 so that the desired thickness could be obtained by controlling the solid content of SU-8 in the final solution. Second, dye-containing SU-8 solutions with different SU-8 solid contents were then spin-coated onto both the dielectric multilayers (BSW platforms), prepared through the deposition (magnetron sputtering system) of five TiO$_2$/SiO$_2$ pairs having respective thicknesses of 71 nm/154 nm on silicon (Si) substrate, and the bare quartz substrates at 3000 rpm for 30 s and baked at 95 °C for 2 min to evaporate the solvent. Third, the Cr photomask with ring patterns was directly attached upside-down onto the R6G-doped SU-8 polymer-coated samples, and then exposed to ultraviolet light having a wavelength of 405 nm using a mask aligner system (MA6 mask aligner, SUSS). After the exposure, the samples were baked at 95 °C for 1.5 min (post-exposure bake), and then developed by a SU-8 developer (1-Methoxy-2-propanol acetate, KAYAKU Advanced Materials) for 60 s, followed by rinsing with isopropyl alcohol (IPA) for 10 s twice and drying in an airflow. The ring cavities having different diameters and ridge widths were then formed on BSW platforms and quartz substrates. Finally, the samples were hard-baked at 150 °C for 5 min to further cross-link the SU-8 polymer and ensure that the properties of SU-8 do not change in actual use under high temperatures.

### Characterizations

UV-Vis absorption spectrum of R6G dye in ethanol/SU-8 solution was measured using a quartz cuvette in a spectrophotometer (V-670, JASCO). The concentration of R6G dye was diluted to 0.2 mM to prevent absorption saturation. The photoluminescence (PL) spectrum of R6G dye in the SU-8 polymer layer was measured using a spectrophotometer (FP-6500, JASCO) under light excitation at the wavelength of 532 nm. The height profiles of the ring cavities were characterized using atomic force microscopy (AFM). The structure and morphology of the R6G-doped SU-8 ring cavities and the cross-sectional image of the dielectric multilayer were characterized using JEM-2800 with a thermal field emission gun at an acceleration voltage of 100 kV.

### Laser performance measurements

The laser performances of the prepared ring cavities on both BSW platforms and quartz substrates were investigated at room temperature using a home-built optical microscopy system, as displayed in Supplementary Fig. 7. A nanosecond (ns) pulsed laser (picolo AOT 1-MOPA, InnoLas Laser) having a wavelength of 532 nm with a 0.6 ns pulse and a repetition rate of 1 kHz was used as the pump light. The pump light was firstly transmitted through a pair of plano-convex lenses, and the spot size of the excitation on the sample surface was adjusted by controlling the distance between these two lenses. Then, the pump light was reflected by the dichroic mirror and focused on the microring cavities using a 20× objective lens with a numerical aperture of 0.45. In the experiment, the diameter of the pump light spot was kept to ~200 μm to ensure uniform illumination. When the ring cavities were optically pumped, their emissions were collected by the same 20× objective lens, and then transmitted through the dichroic mirror and split by a 50:50 beam splitter. On one hand, a tube lens was used to

collect the emission light into a charge-coupled device (CCD) camera (CS505CU, Thorlabs, Inc.) for the optical image observation. On the other hand, a 10× objective lens was used to collect the emission light, and an optical fiber was used to guide the light into the spectrometer (SpectraPro HRS-500, Princeton Instruments) for the spectral analysis. A diffraction grating with a high groove density of 2400 grooves/mm was used. The spectral resolution of this spectrometer system is 0.015 nm, which is among the best for grating-based spectrometers.

## Numerical simulation

The real part and imaginary part of the mode effective indices and the electric field distributions of the guided photonic, hybrid plasmonic, and BSW mode within a SU-8 ridge waveguide were computed as functions of thickness $d$ and ridge width $w$ using the finite-element method (COMSOL Multiphysics, COMSOL Inc.). The confinement factor $\Gamma$, which is defined as the ratio between the electric energy in the SU-8 ridge and the total electric energy of the mode, was calculated using the equation:

$$\Gamma = \frac{\iint_{\text{gain}} W(r) d^2 r}{\iint_{-\infty}^{\infty} W(r) d^2 r}. \tag{4}$$

To investigate the gain mechanism of the proposed BSW lasers, the quality factor ($Q_{\text{leak}}$) related to the lateral radiation leakage due to the effect of curvature and the leakage of light to the air or the substrate was calculated using the results of the eigenfrequency analysis performed on an axisymmetric model implemented in the COMSOL software. For all simulations, perfectly matched layer boundary conditions were used at the edges of the simulated domain.

## Reporting summary

Further information on research design is available in the Nature Portfolio Reporting Summary linked to this article.

## Data availability

The data that support the findings of this study are available from the corresponding author upon request.

## Code availability

The codes used in this work are available from the authors upon request.

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

## Acknowledgements

This work was supported by JSPS KAKENHI (JP20F40045, JP21H01383, JP20H02197). Part of the work was conducted at the Advanced Characterization Nanotechnology Platform of the University of Tokyo, supported by the "Nanotechnology Platform" of the MEXT, Japan (JPMXP09F-21-UT-0021). The authors would like to extend their grateful appreciation for technical support to Dr. Hiroyasu Yamahara and Prof. Hitoshi Tabata from the School of Engineering, The University of Tokyo, and Professor Nicolas Verrier and Professor Olivier Haeberle from Universite de Haute-Alsace. B.-W.L and M.-H.C. acknowledge the fellowship from WINGS-QSTEP.

## Author contributions

Y.-C.L., Y.-L.H., and J.-J.D. conceived the presented idea and designed the work. Y.-C.L. fabricated the samples, performed the experiments, and analyzed the data. Y.-C.L. and B.-W.L. performed sample characterizations. Y.-C.L. and M.-H.C. performed the theoretical simulations. Y.-C.L., J.-J.D. and D.X. built the measurement setup. Y.-L.H., H.D., and J.-J.D. organized the project. Y.-C.L. wrote the paper. Y.-L.H., H.D. and J.-J.D. planned the experiments, supervised the work, and revised the paper.

## Competing interests

The authors declare no competing interests.
