## [Peer Review File · Nature Communications]

High-Q lasing via all-dielectric Bloch-surface-wave platformREVIEWER COMMENTS

Reviewer #1 (Remarks to the Author):

The authors report lasing structures where the lasing mode is a Bloch surface wave supported by a dye-doped polymer rib on a multilayer dielectric Bragg stack. The rib consists of R6G doped SU8. The paper is interesting, well organised, well written, and of high quality, but I do not think it makes the cut for publication in Nature Comms because I think that the paper lacks a significant conceptual advance, and I don't buy the main claim of subwavelength lasing used to motivate publication in this journal.

Regarding the former, dye lasers, dye-doped polymer lasers and dielectric BSWs are all known. Putting them together does not constitute a major conceptual advance in my opinion, but it does constitute interesting and worthwhile work that should be published (elsewhere).

Regarding the latter, the BSW laser proposed is not a subwavelength laser nor is it a nano laser. In such lasers, the *size* of the lasing mode is well below the guided wavelength (thus the term subwavelength) - dielectric BSW waveguides cannot support such modes. This point can be obviously observed from the mode plots of Figs. 1(h) and 1(i) which reveal significant penetration of evanescent fields into the air and multilayer claddings. Inspecting Fig. 1(i) for a core thickness of 80 nm reveals that the $1/e |E|$ decay length is at least two periods into the stack (using the authors' color bar), yielding 450 nm. The mode also decays $1/e |E|$ into the air at least 160 nm. This makes the mode thickness of ~ 610 nm larger than even the free-space wavelength of operation of ~ 575 nm. It is also noted that the mode width is larger than the rib width which is 1500 nm - it is therefore clear that the mode size is not subwavelength along any direction. Indeed, the authors used rib thicknesses of 140 nm and larger, and large radii of curvature ($> 15 \mu\text{m}$), for lasing in their ring structures - subwavelength waveguides are much more strongly confined and able to round bends of much smaller radii.

Also, regarding Fig. 1, confinement factors and effective indices, while useful, do not determine whether a mode is subwavelength - only the mode size can be used to assess this. Furthermore, guidance in the BSW system to rib thicknesses below the cut-off of the SU-8/quartz system does not mean that the diffraction limit has been broken nor does it imply subwavelength guidance - recall that a high-index dielectric core in a homogeneous low-index background supports fundamental TE and TM modes with no cut-off dimensions. Although impractical, an SU-8 core in air should show guiding characteristics similar or better than the BSW rib.

The authors need to remove all claims to a subwavelength laser or to a nano laser throughout the manuscript including in the title. Correspondingly, comparisons with plasmonic nano-lasers should be modulated - the performance of plasmonic nano-lasers clearly still needs much improvement, but usually they have at least one dimension that is incontestably subwavelength (unlike the present work). The more appropriate comparisons are those relative to the doped SU-8/quartz system.

The spectral plot of Fig. 3c is poorly resolved - making the FWHM measurement non-credible. Can the measurement resolution be improved perhaps by using a better spectrograph?

Bleaching is a known problem in dye-doped polymers, and the lasers proposed by the authors are not immune to this problem. Can the authors add some discussion on this point and perhaps show experimental bleaching data?

Reviewer #2 (Remarks to the Author):

In this work, the authors demonstrate for the first time a practical implementation of a

BSW-based, optically-pumped laser device exploiting an organic dye as active material, embedded within a polymeric resonant structure. At my best knowledge, the work and the presented results are novel. Furthermore, this work brings significant advancements in the field of all-dielectric active nanophotonics, expanding the scope of BSW platforms to integrated optical sources.

Briefly, the authors design and fabricate a dye-doped polymeric ring resonator on top of a dielectric multilayer supporting BSW and demonstrate a low-threshold lasing effect coupled to whispering gallery modes which are inherently surface-bound. The advantages of the proposed configuration are highlighted by providing comparisons with other similar geometries, involving optical guided modes and plasmonic modes in dielectric-loaded ridges. Comparisons are performed between computational models and also experimental measurements (for optical modes only) in terms of lasing thresholds. Finally, an assessment of the proposed BSW-laser performance against external perturbations (refractive index changes caused by exposition to solvents) is also conducted.

The work seems rigorously conducted. The manuscript is well-written and the figures are clear, meaningful and easy to interpret.

Still, I have several comments and suggestions worth to be considered by the authors to improve the manuscript quality.

- Bibliography.

The reference list is pertinent and satisfactory but it can be improved. Some hints in the following:

a) At ref.12, when mentioning BSW platforms in the IR region, I suggest to include the following reference 10.1021/acsp Photonics.7b01315, which has been published earlier than ref.12.

b) When mentioning BSW coupling and focusing, I suggest to cite the following work, 10.1038/srep05428, which provides the first demonstration of grating-coupling BSW and their in-plane focusing.

c) At refs. 20-23, I suggest to cite this work 10.1002/lpor.202100542, showing a miniaturized prism for BSW coupling in Otto configuration.

d) When mentioning BSW platforms including polymer stripes doped with emitters (ref. 24), I suggest to include this work: 10.1063/1.3684272, providing the first experimental demonstration of BSW-assisted emission coupling within fluorescent waveguides (dates back to 2012).

e) The idea of BSW ring-resonator is firstly proposed in this work published in 2018 10.1016/j.optcom.2017.10.068. Instead, the first experimental implementation of a BSW resonator is found in this work: 10.1063/1.5093435. I think it should be worth citing both of them.

f) These works deserve to be mentioned because they addresses valuable aspects in the design of BSW waveguides 10.1364/OL.412625 and ring resonators: 10.1364/JOSAB.32.000431

- Technical comments

a) The confinement of BSW in ridges is shown to depend on the ridge thickness and width. However, the positioning of the BSW dispersion within the forbidden band of the multilayer plays an important role, as highlighted in 10.1063/1.5093435. From Figure S2, one could imagine to have a BSW available in the lasing band of the dye even with much smaller thickness of the SU8 ridge, by designing a multilayer with a slightly red-shifted forbidden band. In addition, the degree of confinement of BSW can be generally increased by operating at larger and larger wavevectors, with the additional advantage of being prevented to have Fourier components of the field spread across the air light-line (see paper above). Can the authors elaborate more on the design strategy of the underlying multilayer? Is it possible for the authors to show how losses and Q factors would change in at least one alternative design of the multilayer with a different positioning of the BSW dispersion? Since the lasing performance is shown to be

dramatically sensitive on leakage losses, this aspect is of outmost importance, in view of a further optimization of a BSW-based laser.

b) The ability of a ridge to confine BSW depends on the effective refractive index contrast, which itself depends on both the width and the thickness of the ridge, as shown here: $10.1063/1.3385729$. Can the authors make an estimate of the minimum radius of curvature allowed to have lateral confinement in the ring resonator? This information would integrate the plot in Fig. 5b.

c) Is the Q-factor calculated in Eq.3 exhibiting any dependency on the wavelength at which the (complex) effective index is considered? Can the authors elaborate more on this?

d) After looking at the field distribution in Fig. 5f, it seems that losses would be increased, because of the higher proximity of the BSW mode to the ridge interface. Instead, this approach provides an underestimation of losses. Why the eigenfrequency analysis misses a correct quantification of propagation losses? What is the point in using such a model?

Reviewer #3 (Remarks to the Author):

The manuscript reports on the development of a microlaser based on Bloch surface waves (BSW) by fabrication of polymer waveguides in the form of ring resonators on the surface of a one-dimensional photonic crystal (PC). Lasing manifests itself in the spectral region of Rhodamine 6G gain as a set of peaks typical of whispering gallery modes (WGM) and is observed at a subdiffractive waveguide thickness. As reference samples, the same ring resonators on a quartz substrate are used, for which the absence of lasing is shown at waveguide thicknesses below the diffraction limit. The possibility of sensing with fabricated lasers is shown.

The all-dielectric BSW platform is very flexible in designing nanophotonic devices for sensing and controlling light, operating in arbitrary spectral ranges. An additional attraction is the possibility of using low-index materials such as polymers for BSW control. In this regard, the development of microlasers for the BSW platform, which can be simply printed from polymers using industrial lithography techniques, is a significant and noteworthy task. However, the manuscript contains a number of inaccuracies and unclear results that should be clarified before the manuscript can be published. The following shortcomings can be distinguished:

The main claim that lasing is based on BSW propagating mode is not rigorously proven.

1.1. In proving the presence of BSW lasing, the authors mainly relied on calculations. However, the mode analysis of waveguides looks incomplete, and the results are presented carelessly. What type of modes (TM or TE) are studied for each of the three systems? In waveguides with a width of $1.5 \mu\text{m}$ or more, higher-order modes can exist (for instance, TE₀₁, TE₀₂, etc in BSW waveguides, see [1,2]). However, the manuscript deals only with fundamental modes. Also, the possible excitation of volume waveguide modes of PC [3] or hybrid BSW-waveguide modes under the polymer ridge is not discussed. For example, can the polymer layer together with the top SiO₂ layer of PC be considered as a waveguide core in which the usual photonic mode can be excited (in this case, PC acts as a mirror, see for example [4])?

1.2. The authors experimentally determined the group index for the BSW ring resonators under study, but did not comment on the obtained value. Nevertheless, the group index allows us to estimate the effective refractive index of the supposedly excited mode if we know its dispersion (for example, from calculations). Considering the small dispersion of the materials used, one can assume that a PC volume waveguide mode with an effective refractive index close to the group index (1.82) is excited in the resonator (such modes are usually easily observed inside a PC in calculations). An appropriate analysis should be performed to refute (or confirm) this assumption. It also makes sense to use the

found group index for WGM photonic resonators in the analysis.

1.3. The BSW platform provides a unique opportunity to study the properties of BSW modes by analyzing the leakage radiation. Since the effective refractive index of BSW modes is lower than the refractive index of PC materials, in the case of transparent substrate, the leakage radiation can be collected using conventional immersion objective lenses [2, 5, 6] or prisms in the Kretschmann scheme [7,8]. This makes it possible not only to determine the mode composition of the BSW waveguides and effective refractive indices of modes, but also to directly visualize the BSW modes [2], which would help to clearly demonstrate the operation of the BSW laser, thereby significantly improving the manuscript.

2. The applicability of developed BSW lasers is discussed too briefly.

2.1. The authors state that the BSW lasers possess a low lasing threshold. Could the authors compare this value with lasing thresholds of others on-chip WGM lasers?

2.2. In Ref. 26, the efficiency of coupling laser radiation into the BSW modes is estimated to be at least 15%, while the lasing threshold is 20 times lower at a smaller laser size. Could the authors estimate the efficiency of coupling radiation from their ring BSW laser into BSW modes, for example, of a straight waveguide and compare it with the case from Ref. 26? Demonstrating the possibility of using the BSW laser for integrated photonics, at least through simulation, would improve the manuscript.

2.3. How does the sensing performance of the BSW laser compare with other WGM lasers used in sensing [9]?

3. Some of the results are presented carelessly. Not all methods are described in sufficient detail to reproduce the work.

3.1. Lines 136-137: «Mode analysis (see details in the Methods section) is first employed to design ... BSW mode». I didn't find in the manuscript how mode analysis is applied to design BSW laser. Moreover, the PC design method is also not fully outlined in Supporting Information. In particular, what angle theta was taken? Why does the PC consist of only 5 pairs of layers? Increasing the number of layers should reduce radiation losses [10], which can improve the lasing properties. In general, the optimization of radiation losses in BSW waveguides is not a trivial task [11, 12].

3.2. Lines 262-263: «Due to the transparent nature of the SU-8 polymer, the Q_{abs} of the ring cavities is of the order of 10^8 [Ref. 45]». In Ref. 45 there is nothing about SU-8, only about PMMA. Moreover, as follows from Figure S1, the absorption of R6G/ethanol/SU-8 solution is not zero inside the lasing band.

3.3. How was eigenfrequency analysis performed, and which axisymmetric model was used?

3.4. Lines 316-318: «... implying that more light can interact with the edge or surface roughness (see Figure S9b-d in the Supporting Information)». Such a statement can hardly be made on the basis of Figure S9b-d.

3.5. Lines 439-440: «To investigate the gain mechanism of the proposed BSW lasers, the radiation-limited quality factor Q_{rad} was calculated...». What is the the radiation-limited quality factor Q_{rad} ?

3.6. The pump energy density scale is confusing in Fig. 3b.

[1] Sfez, T. et al. Bloch surface waves in ultrathin waveguides: near-field investigation of mode polarization and propagation. J. Opt. Soc. Am. B 27, 1617 (2010).

- [2] Safronov, K. R. et al. Multimode Interference of Bloch Surface Electromagnetic Waves. *ACS Nano* 14, 10428–10437 (2020).
- [3] Yeh, P., Yariv, A., & Hong, C. S. Electromagnetic propagation in periodic stratified media. I. General theory. *JOSA*, 67(4), 423-438 (1977).
- [4] Liscidini, M. Surface guided modes in photonic crystal ridges: the good, the bad, and the ugly. *J. Opt. Soc. Am. B* 29, 2103 (2012).
- [5] Descrovi, E. et al. Leakage radiation interference microscopy. *Opt. Lett.* 38, 3374 (2013).
- [6] Safronov, K. R. et al. Miniature Otto Prism Coupler for Integrated Photonics. *Laser and Photon. Rev.* 16, 2100542 (2022).
- [7] Moskalenko, V.V. et al. Surface wave-induced enhancement of the Goos-Hänchen effect in one-dimensional photonic crystals. *Jetp Lett.* 91, 382–386 (2010).
- [8] Pidgayko, et al. Direct imaging of isofrequency contours of guided modes in extremely anisotropic all-dielectric metasurface. *ACS Photonics*, 6(2), 510-515 (2018).
- [9] Toropov, N., Cabello, G., Serrano, M.P. et al. Review of biosensing with whispering-gallery mode lasers. *Light Sci Appl* 10, 42 (2021).
- [10] Vosoughi Lahijani, B. et al. Centimeter-Scale Propagation of Optical Surface Waves at Visible Wavelengths. *Advanced Optical Materials*, 2102854 (2022).
- [11] Perani, T. & Liscidini, M. Long-range Bloch surface waves in photonic crystal ridges. *Opt. Lett.* 45, 6534 (2020).
- [12] Luo, H., Tang, X., Lu, Y., & Wang, P. Low-Loss Photonic Integrated Elements Based on Bound Bloch Surface Wave in the Continuum. *Physical Review Applied*, 16(1), 014064 (2021). 
List of important revisions

Additional experimental data and discussions

1. We have investigated the optical characteristics of the lasing mode of the BSW lasers by leakage radiation microscopy and added the section **“Investigation into the BSW lasing mode by leakage radiation microscopy”** and Figure S13 in the Supporting Information. **These experimental results show that the effective index of the lasing mode corresponds to that of the theoretical BSW mode.** (Response to Reviewer 3’s comments)
2. We have added a discussion section **“Discussion of the group index of the BSW lasers”** and Figure S9 in the Supporting Information. Here, **we show by computing dispersion diagrams that the photonic mode and BSW mode possess different values for their group indices and confirm experimentally these values.** The lasing action of the BSW lasers is therefore based on the stimulated emission of the BSW mode. The manuscript has been revised accordingly. (Response to Reviewer 3’s comments)
3. We have investigated the effect of the design of the BSW multilayer platform on the BSW lasing behavior and added a section **“Investigation of multilayer design for the BSW laser”** and Figures S17 and S18 in the Supporting Information. **These new experimental results show improved lasing performance (decreased lasing threshold) for smaller laser cavities.** We have also revised the manuscript text accordingly. (Response to Reviewer 2’s comments)
4. We have added a section **“Photobleaching of the BSW lasers made of R6G-doped SU-8 polymer”** and Figure S20 in the Supporting Information. Also, we have revised the manuscript text accordingly. (Response to Reviewer 1’s comments)

5. We have added Figure S8 in the Supporting Information to clarify the **spectral resolution of our measurements**. (Response to Reviewer 1's comments)

Additional simulations and discussions

1. We have revised the manuscript by adding a discussion about the **effect of the cavity diameter**. We have also **revised Figure 5b** and the corresponding figure caption to integrate information about the Q_{leak} of the BSW ring cavities with the small diameters D of 10, 15, and 20 μm . We have revised the manuscript text accordingly. (Response to Reviewer 2's comments)
2. We have added statements in the manuscript and revised Figures 5(b,d,f), Figure S14, and Figure S21 to present our **numerical investigation into the loss mechanisms of the BSW lasers and the environmental sensitivity of the fabricated BSW platform** (the actual layer thicknesses were slightly different from the designed thicknesses according to cross-sectional SEM image analysis of the sample).
3. We have added Figure S14 and the corresponding figure caption to explain the **dependency of the quality factor on the wavelength using the loss estimated by Equation 3** from mode analyses. The manuscript text was revised accordingly. (Response to Reviewer 2's comments)
4. We have added Figure S15 to include **simulated results about the dependency of the surface scattering losses with the ridge width**. We have also revised the discussion part related to losses and deleted the statement about the underestimation of propagation losses in the eigenfrequency analysis. (Response to Reviewer 2's comments)

5. We have added a section discussing the **simulated results for the coupling between a BSW ring laser cavity and a waveguide** entitled “Coupling between a BSW ring laser cavity and a waveguide” and Figure S19 in the Supporting Information. The manuscript has been revised accordingly. (Response to Reviewer 3’s comments)
6. We have comprehensively investigated the **high-order modes** (Figure S11) and **internal modes** (Figure S12) sustained by the proposed BSW laser structure. We have added a discussion section “Detailed discussions about the verification of the BSW lasing” and Figures S11 and S12 in the Supporting Information. (Response to Reviewer 3’s comments)

Other revisions for the discussions and references

1. We have added Table S2 in the Supporting Information to **compare our BSW laser with other on-chip WGM lasers based on dye-doped polymers**. (Response to Reviewer 3’s comments)
2. We have indicated in the manuscript that the **photonic mode is a transverse electric (TE) mode, the BSW mode is a TE-like guided mode, and the hybrid plasmonic mode is a transverse magnetic (TM) mode**. (Response to Reviewer 3’s comments)
3. We have corrected the mistake on the quality factor related to lateral radiation leakage (renamed Q_{rad} into Q_{leak}). (Response to Reviewer 3’s comments)
4. We have revised the manuscript to give a detailed description of the employed methods for simulation. (Response to Reviewer 3’s comments)

5. We have deleted all occurrences of “nano laser” and “subwavelength” in the manuscript (including in the title) as well as correctly reformulated the description of the mode. (Response to Reviewer 1’s comments)
6. We have corrected the pump energy density scale in Figure 3b. (Response to Reviewer 3’s comments)
7. We have added references 12, 16, 19, 21–23, 30, and 31 in the manuscript. We have also added descriptions of the cited works. (Response to Reviewer 2’s comments)
8. We have corrected the reference related to the optical constants of the SU-8 polymer (MicroChem. SU-8 2000 Permanent epoxy negative photoresist: Processing guidelines for SU-8 2000.5, SU-8 2002, SU-8 2005, SU-8 2007, SU-8 2010 and SU-8 2015. Available from: <http://www.microchem.com/pdf/SU-82000DataSheet2025thru2075Ver4.pdf> (2017).). (Response to Reviewer 3’s comments)

Point-to-point responses to Reviewers' comments

Reviewer #1 (Remarks to the Author):

The authors report lasing structures where the lasing mode is a Bloch surface wave supported by a dye-doped polymer rib on a multilayer dielectric Bragg stack. The rib consists of R6G doped SU8. The paper is interesting, well organized, well written, and of high quality, but I do not think it makes the cut for publication in Nature Comms because I think that the paper lacks a significant conceptual advance, and I don't buy the main claim of subwavelength lasing used to motivate publication in this journal.

Regarding the former, dye lasers, dye-doped polymer lasers and dielectric BSWs are all known. Putting them together does not constitute a major conceptual advance in my opinion, but it does constitute interesting and worthwhile work that should be published (elsewhere).

We agree that dye lasers, dye-doped polymer lasers, and dielectric BSWs have been reported. Indeed, the optical properties of BSWs have been well-investigated, and several types of BSW-based two-dimensional (2D) optical devices have been developed in past studies, indicating the potential applications of BSWs in on-chip integrated systems.

However, a coherent light source that is based on the stimulated emission of the BSW propagating mode has not been reported. In this study, we propose and demonstrate an all-dielectric BSW laser for the first time. This BSW laser can be integrated into an optical circuit on a multilayer BSW platform in an efficient manner due to the mode nature having low propagation losses and small dimensions enabling

miniaturization of the proposed light source, and the availability of BSW-based components such as mode guiding structures and modulators.

Therefore, we believe that the proposed BSW laser is the key component to complete the whole set of required components for on-chip BSW-based devices or systems, and the present work should help to pave the way for future applications of BSWs in integrated optical circuits.

Regarding the latter, the BSW laser proposed is not a subwavelength laser nor is it a nano laser. In such lasers, the *size* of the lasing mode is well below the guided wavelength (thus the term subwavelength) - dielectric BSW waveguides cannot support such modes. This point can be obviously observed from the mode plots of Figs. 1(h) and 1(i) which reveal significant penetration of evanescent fields into the air and multilayer claddings. Inspecting Fig. 1(i) for a core thickness of 80 nm reveals that the $1/e$ $|E|$ decay length is at least two periods into the stack (using the authors' color bar), yielding 450 nm. The mode also decays $1/e$ $|E|$ into the air at least 160 nm. This makes the mode thickness of ~ 610 nm larger than even the free-space wavelength of operation of ~ 575 nm. It is also noted that the mode width is larger than the rib width which is 1500 nm – it is therefore clear that the mode size is not subwavelength along any direction. Indeed, the authors used rib thicknesses of 140 nm and larger, and large radii of curvature (> 15 μm), for lasing in their ring structures – subwavelength waveguides are much more strongly confined and able to round bends of much smaller radii.

Also, regarding Fig. 1, confinement factors and effective indices, while useful, do not determine whether a mode is subwavelength - only the mode size can be used to assess this. Furthermore, guidance in the BSW system to rib thicknesses below the cut-off of the SU-8/quartz system does not mean that the diffraction limit has been broken nor does it imply subwavelength guidance – recall that a high-index dielectric core in a homogeneous low-index background supports fundamental TE and TM modes with no cut-off dimensions. Although impractical, an SU-8 core in air should show guiding characteristics similar or better than the BSW rib.

The authors need to remove all claims to a subwavelength laser or to a nano laser throughout the manuscript including in the title. Correspondingly, comparisons

with plasmonic nano-lasers should be modulated – the performance of plasmonic nano-lasers clearly still needs much improvement, but usually they have at least one dimension that is incontestably subwavelength (unlike the present work). The more appropriate comparisons are those relative to the doped SU-8/quartz system.

Thank you for pointing out this issue. We agree that the subwavelength claim is not appropriate considering that the reported mode size is comparable to the wavelength, and we have deleted all occurrences of “nano laser” and “subwavelength” in the manuscript (including in the title) as well as correctly reformulated the description of the mode.

Regarding the comparison between BSWs and surface plasmon polaritons, we feel that the comparison should still be of interest to the community involved in surface wave-related topics because BSW is usually considered a dielectric analog of surface plasmon polaritons.

The spectral plot of Fig. 3c is poorly resolved – making the FWHM measurement non-credible. Can the measurement resolution be improved perhaps by using a better spectrograph?

We have added **Figure S8** in the Supporting Information to clarify the effect of the spectral resolution in our measurements and revised the manuscript accordingly.

In this study, we used a spectrometer system with a grating having 2400 grooves per mm to measure the emission spectra of the BSW lasers. The spectral resolution of this spectrometer system is 0.015 nm and among the best for grating-based spectrometers. The smallest reported FWHM is 0.019 nm and obtained from the BSW laser having a diameter of 50 μm . This result is therefore limited by the spectrometer system resolution, implying that the actual smallest FWHM could be smaller than this value. Note that this case corresponds to the best-performing BSW lasers. For many fabricated BSW lasers, the FWHMs of the lasing peaks are slightly larger than 0.02 nm, as shown in Figure S8. These peaks are indeed resolved (the lasing peak has more than one data point), suggesting that the current measurement system is good enough to obtain FWHM in our work. Although, as the referee pointed out, our best device may not be accurately characterized in terms of FWHM, this spectral resolution limitation implies that only the reported FWHM value for the best device may be overestimated and the conclusions of the work, therefore, remain the same.

Figure S8. The magnified emission spectra of BSW lasers having (a) $d = 136$ nm, $w = 1.5$ μm and $D = 50$ μm , and (b) $d = 136$ nm, $w = 1.5$ μm , and $D = 30$ μm . Note that the FWHMs are slightly larger than the spectral resolution of the spectrometer system.

The revised part of the manuscript is provided as follows: (In *Methods* section)

“A diffraction grating with a high groove density of 2400 grooves/mm was used. The spectral resolution of this spectrometer system was 0.015 nm, which is among the best for grating-based spectrometers.”

Bleaching is a known problem in dye-doped polymers, and the lasers proposed by the authors are not immune to this problem. Can the authors add some discussion on this point and perhaps show experimental bleaching data?

We have added a section related to this issue entitled “*Photobleaching of the BSW lasers made of R6G-doped SU-8 polymer*” and **Figure S20** in the Supporting Information and revised the manuscript accordingly.

We agree that the proposed BSW laser consisting of a rhodamin 6G (R6G)-doped SU-8 ring cavity should suffer from photobleaching, which is a common issue for dye molecules when exposed to intense light. Thus, we have investigated the photobleaching of the BSW lasers by continuously pumping laser devices and collecting its emission intensity. Figure S20 shows the normalized emission intensity of a BSW laser ($D = 100 \mu\text{m}$) as a function of the number of pump pulses. The emission intensity decreases to half of its initial value after approximately 20,000 pulses. Note that this lifetime is of a similar order of magnitude as that reported for R6G-doped SU-8 polymer lasers^{1,2}. Although the photobleaching of dye-doped polymer restricts the lifetime of the proposed BSW laser, the fabrication method is simple, cheap, and rapid, making it suitable for large production. Furthermore, the photobleaching can be reduced by operating the laser at low temperature³, working under an inert atmosphere³, or filling the free volume in the polymer host with additives (such as diphenyl thiourea⁴), and so the photostability of such BSW lasers can still be improved to obtain more favorable stability for long-term applications. We also note that the R6G-doped SU-8 polymer was chosen as the gain medium in this study to provide a proof of concept for the BSW laser. We believe that the applicability of the proposed BSW laser can be drastically extended by employing an inorganic gain medium (e.g., Er-doped Si and GaN) so that the biosensing

applications of the BSW lasers will be made possible.

Figure S20. The normalized emission intensity of the BSW laser as a function of the number of pump pulses. The diameter of the BSW ring laser was 100 μm , and the pump energy density 10.9 $\mu\text{J}/\text{mm}^2$.

The revised part of the manuscript is provided as follows: (Line 2 page 19)

“Before performing the sensitivity study, the stability of the BSW laser is inspected (regarding the photobleaching effect, see Figure S20 in the Supporting Information).”

References

1. Balslev, S., Rasmussen, T., Shi, P., Kristensen, A. Single mode solid state distributed feedback dye laser fabricated by gray scale electron beam lithography on a dye doped SU-8 resist. *J. Micromech. Microeng.* **15** 2456–2460 (2005).
2. Nilsson, D., Balslev, S., Gregersen, M. M., Kristensen, A. Microfabricated solid-state dye lasers based on a photodefinable polymer. *Appl. Opt.* **44**, 4965 (2005).
3. Zondervan, R., Kulzer, F., Kol’chenko, M. A., Orrit, M. Photobleaching of Rhodamine 6G in poly(vinyl alcohol) at the ensemble and single-molecule levels. *J. Phys. Chem. A* **108**, 1657–1665 (2004).
4. Singh, S., Kanetkar, V. R., Sridhar, G., Muthuswamy, Raja, K. Solid-state polymeric dye lasers. *J. Lumin.* **101**, 285–291 (2003).

Reviewer #2 (Remarks to the Author):

In this work, the authors demonstrate for the first time a practical implementation of a BSW-based, optically-pumped laser device exploiting an organic dye as active material, embedded within a polymeric resonant structure. At my best knowledge, the work and the presented results are novel. Furthermore, this work brings significant advancements in the field of all-dielectric active nanophotonics, expanding the scope of BSW platforms to integrated optical sources.

Briefly, the authors design and fabricate a dye-doped polymeric ring resonator on top of a dielectric multilayer supporting BSW and demonstrate a low-threshold lasing effect coupled to whispering gallery modes which are inherently surface-bound. The advantages of the proposed configuration are highlighted by providing comparisons with other similar geometries, involving optical guided modes and plasmonic modes in dielectric-loaded ridges. Comparisons are performed between computational models and also experimental measurements (for optical modes only) in terms of lasing thresholds. Finally, an assessment of the proposed BSW-laser performance against external perturbations (refractive index changes caused by exposition to solvents) is also conducted.

The work seems rigorously conducted. The manuscript is well-written and the figures are clear, meaningful and easy to interpret.

Still, I have several comments and suggestions worth to be considered by the authors to improve the manuscript quality.

- Bibliography.

The reference list is pertinent and satisfactory but it can be improved. Some hints in the following:

a) At ref.12, when mentioning BSW platforms in the IR region, I suggest to include the following reference 10.1021/acsphotonics.7b01315, which has been published earlier than ref.12.

b) When mentioning BSW coupling and focusing, I suggest to cite the following work, 10.1038/srep05428, which provides the first demonstration of grating-coupling BSW and their in-plane focusing.

c) At refs. 20-23, I suggest to cite this work 10.1002/lpor.202100542, showing a miniaturized prism for BSW coupling in Otto configuration.

d) When mentioning BSW platforms including polymer stripes doped with emitters (ref. 24), I suggest to include this work: 10.1063/1.3684272, providing the first experimental demonstration of BSW-assisted emission coupling within fluorescent waveguides (dates back to 2012).

e) The idea of BSW ring-resonator is firstly proposed in this work published in 2018 10.1016/j.optcom.2017.10.068. Instead, the first experimental implementation of a BSW resonator is found in this work: 10.1063/1.5093435. I think it should be worth citing both of them.

f) These works deserve to be mentioned because they addresses valuable aspects in the design of BSW waveguides 10.1364/OL.412625 and ring resonators: 10.1364/JOSAB.32.000431

Indeed the above references improve our Reference list, and so we have added them to our manuscript as references 12, 16, 19, 21–23, 30, and 31. We have also added some descriptions of the cited works as follows:

(Line 5 page 5) “*A circular grating structure has been proposed to demonstrate BSW coupling and focusing¹⁹. Two-dimensional (2D) disk resonator²⁰, as well as ring resonator^{21–23} sustaining BSWs have been designed on the dielectric multilayer, and the optical resonance of BSWs has been realized experimentally^{20,23}.*”

(Line 1 page 6) “*(e.g., fluorescent dye doped in polymer¹¹, fluorescent proteins grafted onto polymeric waveguide³¹, and quantum dots³²)*”.

- Technical comments

a) The confinement of BSW in ridges is shown to depend on the ridge thickness and width. However, the positioning of the BSW dispersion within the forbidden band of the multilayer plays an important role, as highlighted in 10.1063/1.5093435. From Figure S2, one could imagine to have a BSW available in the lasing band of the dye even with much smaller thickness of the SU8 ridge, by designing a multilayer with a slightly red-shifted forbidden band. In addition, the degree of confinement of BSW can be generally increased by operating at larger and larger wavevectors, with the additional advantage of being prevented to have Fourier components of the field spread across the air light-line (see paper above). Can the authors elaborate more on the design strategy of the underlying multilayer? Is it possible for the authors to show how losses and Q factors would change in at least one alternative design of the multilayer with a different positioning of the BSW dispersion? Since the lasing performance is shown to be dramatically sensitive on leakage losses, this aspect is of outmost importance, in view of a further optimization of a BSW-based laser.

We have added a section related to this issue entitled “*Investigation of multilayer design for the BSW laser*” and Figure S17 and S18 in the Supporting Information. (We have also renamed $Q_{\text{leak-prop}}$ into $Q_{\text{leak-surr}}$ to avoid confusion, as $Q_{\text{leak-surr}}$ represents the leakage to the surrounding including leakages through the bottom substrate and ridge sides.)

As noted by the reviewer, the BSW mode wavevector will become larger when designing the multilayer for a red-shifted forbidden band. Also, a BSW mode in the lasing band could still be sustained within a SU-8 polymer ridge waveguide having a

smaller thickness by increasing the degree of confinement of the BSW mode. To investigate this behavior, we calculated the BSW dispersion diagram of the TiO₂/SiO₂ multilayer having the respective thicknesses of 71 nm/154 nm (original design) for a SU-8 layer on the multilayer top including the case of a smaller thickness of 100 nm (Figure S17a). For comparison purposes, the BSW dispersion diagrams for the red-shifted forbidden band with different TiO₂/SiO₂ multilayers (two additional designs with thicknesses of 78 nm/170 nm and 89 nm/194 nm, which are designed at wavelengths of 640 and 730 nm, respectively, based on the equation S1) having a top 100-nm-thick SU-8 layer were also computed (Figure S17b). In Figure S17a, a smaller thickness of the SU-8 polymer layer results in less optical confinement (as displayed in Figures 1a–c in the manuscript), and the dispersion diagram exhibits smaller wavevectors k_x . As shown in Figure S17b, when the forbidden band of the TiO₂/SiO₂ multilayer redshifts, the dispersion of BSW modes red-shifts simultaneously, implying larger wavevectors k_x for the BSW modes in the lasing band (at approximately 580 nm) of the R6G-doped SU-8 ring lasers (see inset of Figure S17b). To further characterize the optical properties of these BSW modes, we calculated (Figures S17c–e) using mode analysis the mode effective indices, the confinement factors, and the electric field norm distributions of these BSW modes sustained within the SU-8 polymer ridge ($d = 100$ nm, $w = 2.5$ μ m) lying onto the surface of different TiO₂/SiO₂ multilayers. It is found that, when red-shifting the forbidden band of the designed multilayer, the $n_{\text{effective}}$ increases while the $k_{\text{effective}}$ decreases, suggesting a better-supported BSW mode as expected from the dispersion diagram. However, the confinement factor (defined as the ratio between the electric energy in the SU-8 ridge and the total electric energy of the mode) decreases when the forbidden band of the designed multilayer was red-shifted. We attribute this behavior to a broadening of the fields in the top SiO₂ layer underneath the SU-8

polymer ridge (see Figures S17 c–e), which will likely result in decreased optical feedback provided for the WGM resonance, especially for small ring diameters resulting in a significant curvature effect.

To clarify possible improvement in the BSW laser design, we investigated the quality factors of the BSW ring cavities ($d = 100$ nm, $w = 2.5$ μm) for very large diameters by using 2D guiding structures (for which the curvature effect is neglected), and ring cavities with a relatively small diameter of $D = 30$ μm onto the surface of different BSW platforms. For the case of very large ring diameters, the effect of the curvature can be neglected so that the leakage-related quality factor Q_{leak} can be estimated by only considering the quality factor $Q_{\text{leak-surr}}$, which is related to the leakage of light to the surrounding (Figure S17f). For the small diameter, the effect of curvature should be taken into account so that the quality factor (Q_{leak}) is evaluated using the eigenfrequency analysis for an axisymmetry model (Figure S17g). As expected, the Q_{leak} values for the BSW ring cavities with large diameters increase with red-shifting the forbidden band of the designed multilayer due to smaller $k_{\text{effective}}$. However, the Q_{leak} values for the BSW ring cavities with a small diameter show a maximum when the BSW platform consists of five $\text{TiO}_2/\text{SiO}_2$ pairs having respective thicknesses of 78 nm/170 nm, which is attributed to a tradeoff between the decrease in $k_{\text{effective}}$ and confinement factor. In summary, the quality factor of the BSW ring cavity can indeed be improved by red-shifting the forbidden band of the multilayer, thus suggesting an improved lasing performance even when the thickness of the SU-8 ridge becomes small.

Figure S17. Investigation into the multilayer design of the BSW laser with a smaller thickness of the SU-8 polymer layer. (a) The BSW dispersion diagrams of the TiO₂/SiO₂ multilayer, having respective thicknesses of 71 and 154 nm with a 136-nm-thick (black) and a 100-nm-thick (blue) top SU-8 polymer layer. (b) The BSW dispersion diagrams of the TiO₂/SiO₂ multilayers having respective thicknesses of 71 nm/154 nm (blue), 78 nm/170 nm (red), and 89 nm/194 nm (green) with a top 100-nm-thick SU-8 layer. The forbidden bands of the TiO₂/SiO₂ multilayers are displayed, and the interested wavelength (580 nm) corresponding to the center of the lasing band in the R6G-doped SU-8 ring lasers is also indicated. Inset in figure b: The magnified graph of the dispersion diagrams showing larger wavevector k_x when designing the multilayer for a red-shifted forbidden band. (c–e) The calculated electric field norm distributions of the BSW modes sustained within the SU-8 polymer ridge ($d = 100$ nm, $w = 2.5$ μm with non-vertical sidewall) onto the surface of TiO₂/SiO₂ multilayers having respective

thicknesses of (c) 71 nm/154 nm, (d) 78 nm/170 nm, and (e) 89 nm/194 nm using mode analysis. The real and imaginary parts of the mode effective indices (expressed by $n_{\text{effective}} + ik_{\text{effective}}$) and confinement factor (CF) are indicated in the figure. (f, g) The simulated Q_{leak} of the BSW ring cavities ($d = 100$ nm, $w = 2.5$ μm) for (f) very large diameters (for which the curvature effect can be neglected) and (g) a small diameter of $D = 30$ μm onto the surface of different multilayers using the eigenfrequency analysis for an axisymmetry model.

In the following, we apply the above strategy to fabricate BSW lasers with improved performance. At this time, as we are not able to fabricate BSW ring lasers with smaller thicknesses, we investigated the effect of the multilayer design on the BSW ring lasers with a thickness of 136 nm and show improved performance in terms of the lasing threshold for small ring diameters. Figure S18a shows the calculated BSW dispersion diagram of the TiO₂/SiO₂ multilayers having respective thicknesses of 71 nm/154 nm (original design) and 89 nm/194 nm, both with a SU-8 layer thickness of 136 nm. As seen in the previous section, the wavevector k_x for the BSW modes increases with red-shifting the forbidden band of the multilayer, and an increase in $n_{\text{effective}}$ and a decrease in $k_{\text{effective}}$ are observed (see Figures S18b, c). Note that the value of the confinement factors slightly decreases, indicating that this multilayer design should benefit the BSW ring lasers even for small ring diameters.

To clarify the possible improvement, we calculated the Q_{leak} of the BSW ring cavities ($d = 136$ nm, $w = 2.5$ μm , $D = 30$ μm) onto the surface of multilayers designed with different forbidden bands using the eigenfrequency analysis for an axisymmetric model, as shown in Figure S18d. The Q_{leak} value is found to be optimized ($Q_{\text{leak}} \sim 16,000$) when the multilayer consists of five TiO₂/SiO₂ pairs having respective thicknesses of 89 nm/194 nm. Next, we fabricated the BSW ring cavities having diameters D in the range between 10 to 100 μm (thickness $d = 136$ nm, width $w = 2.5$

μm) onto the surface of the improved BSW platform (consisting of five pairs of TiO_2 and SiO_2 layers having respective thicknesses of 89 nm/194 nm). Then the fabricated BSW lasers were optically pumped with a nanosecond laser to examine their lasing performances (Figure S18e). The measured lasing threshold values of the BSW lasers having the large diameters of 50, 75, and 100 μm are approximately the same as the BSW lasers onto the original design multilayer (Figure 5a in the manuscript), implying that the lasing thresholds of the revised design should be limited by other losses than the leakage-related losses (losses from the effect of curvature and propagation) such as the surface scattering losses. Importantly, we found that the lasing threshold value is significantly reduced to 33% of that of the original design for the BSW laser with a diameter of 30 μm , which is attributed to the decrease in leakage-related losses. Furthermore, the minimum diameter achieving lasing is shrunk to 20 μm (Figure S18e). According to these results, we confirmed that the lasing performance of the BSW lasers could be improved using a better design of the BSW platform.

Figure S18. Improved lasing performance for the BSW lasers in terms of the multilayer design. (a) The BSW dispersion diagrams of the $\text{TiO}_2/\text{SiO}_2$ multilayers, having respective thicknesses of 71 nm/154 nm (blue) and 89 nm/194 nm (red), both with a 136-nm-thick top SU-8 polymer layer. (b, c) The calculated electric field norm distributions of the BSW modes sustained within the SU-8 polymer ridge ($d = 136$ nm, $w = 2.5 \mu\text{m}$ with non-vertical sidewall) onto the surface of $\text{TiO}_2/\text{SiO}_2$ multilayers having respective thicknesses of (b) 71 nm/154 nm and (c) 89 nm/194 nm. The real and imaginary parts of the mode effective indices (expressed by $n_{\text{effective}} + ik_{\text{effective}}$) and confinement factor (CF) are indicated in the figure. (d) The simulated Q_{leak} of the BSW ring cavities ($d = 136$ nm, $w = 2.5 \mu\text{m}$, $D = 30 \mu\text{m}$) onto the surface of different multilayers using the eigenfrequency analysis for an axisymmetry model. (e) The measured lasing threshold values of the BSW ring lasers ($d = 136$ nm, $w = 2.5 \mu\text{m}$) onto the surface of the original (blue circle) and improved (green ring) BSW platforms.

The corresponding revised part of the manuscript is provided as follows: (Line 10 page 18) “So far, we have investigated the loss mechanisms of the BSW lasers,

which are found to strongly depend on the optical confinement of the BSW guided mode. It is also known that losses can be decreased using a multilayer design with a red-shifted forbidden band¹⁶, a strategy employed to obtain improved lasing performance (see Figs. S17 and S18)."

b) The ability of a ridge to confine BSW depends on the effective refractive index contrast, which itself depends on both the width and the thickness of the ridge, as shown here: 10.1063/1.3385729. Can the authors make an estimate of the minimum radius of curvature allowed to have lateral confinement in the ring resonator? This information would integrate the plot in Fig. 5b.

We have revised the manuscript by adding a **discussion about the effect of the cavity diameter**. We have also revised **Figure 5b** and the corresponding figure caption to integrate information about the Q_{leak} of the BSW ring cavities with the diameters D of 10, 15, and 20 μm .

The BSW ring cavities having diameters D of 10, 15, and 20 μm (thickness $d = 136$ nm, width $w = 1.5, 2,$ and 2.5 μm) were also fabricated and optically pumped by a nanosecond laser, however, these ring cavities did not lase. To investigate their losses, the leakage-related quality factors Q_{leak} have been simulated and the results added in Figure 5b. The Q_{leak} values of the BSW ring cavities with $D = 10, 15,$ and 20 μm decrease with decreasing diameter ($Q_{\text{leak}} \sim 800\text{--}3,000$), implying larger lateral radiation leakage due to the effect of the curvature. Even the Q_{leak} values of the ring cavities with $D = 15$ and 20 μm ($Q_{\text{leak}} \sim 2,000$ and $3,300$) are higher than those of the photonic ring cavities ($Q_{\text{leak}} \sim 1200$), lasing could not be observed which suggest the presence of other losses limiting the lasing performance (e.g., scattering losses; see additional discussion on losses in Reviewer 3's replies). From the above discussion, we conclude that the minimum diameter of the BSW ring cavities that can lase should be in the range of 20 to 30 μm for the current multilayer design (five pairs of $\text{TiO}_2/\text{SiO}_2$ layers having respective thicknesses of 71 nm/154 nm). While lasing could be observed from the BSW ring cavity having a smaller diameter $D = 20$ μm for the improved multilayer design (five pairs of $\text{TiO}_2/\text{SiO}_2$ layers having respective

thicknesses of 89 nm/194 nm) as discussed in the previous reply.

Figure 5. Lasing threshold investigation of BSW lasers. (a) The measured lasing threshold values of BSW lasers. The thresholds of two photonic lasers are also displayed. Note that the BSW ring cavities having diameters of 10, 15, and 20 μm did not lase. (b) The simulated leakage-related quality factor (defined as Q_{leak}) of the ring laser cavities. (c–f) The calculated electric field norm distributions for the photonic ring lasers with (c) $d = 240$ nm and $w = 1.5$ μm, and (e) $d = 240$ nm and $w = 2.5$ μm, and for BSW ring lasers with (d) $d = 136$ nm and $w = 1.5$ μm, and (f) $d = 136$ nm and $w = 2.5$ μm obtained by eigenfrequency analysis of an axisymmetric model. The diameters of the ring cavities are 100 μm.

In the following, we present the revisions of our manuscript according to the above discussion:

(Line 1 page 11) “*The ring cavities were designed and fabricated with different thicknesses ($d = 140$ to 500 nm), diameters ($D = 10$ to 100 μm), and ridge widths ($w = 1.5$ to 2.5 μm), as shown in Figure 2b.*”

(Line 10 page 12) *“Figure 4a displays the lasing spectrum of the BSW lasers having different diameters ranging from 30 to 100 μm . Note that the BSW ring cavities having diameters of 10, 15, and 20 μm did not lase.”*

(Line 16 page 15) *“Note that although the Q_{leak} values of the ring cavities with $D = 15$ and 20 μm ($Q_{\text{leak}} \sim 2,000$ and 3,300) are higher than those of the photonic ring cavities ($Q_{\text{leak}} \sim 1,200$), lasing could not be observed thus suggesting the presence of other losses (e.g., scattering losses) limiting the lasing performance as discussed in the later section.”*

c) Is the Q-factor calculated in Eq.3 exhibiting any dependency on the wavelength at which the (complex) effective index is considered? Can the authors elaborate more on this?

We have added **Figure S14** and the corresponding figure caption to explain the dependency of the quality factor on the wavelength using the loss estimated using Equation 3 from mode analyses. The manuscript text was revised accordingly.

The calculations of $Q_{\text{leak-surr}}$ given by Equation 3 were performed by mode analysis at 580 nm, which corresponds to the wavelength roughly centered at the lasing band observed in the R6G-doped SU-8 ring lasers. To understand the wavelength dependence of $Q_{\text{leak-surr}}$, we have graphed the real part (revised Figure S14a) and imaginary part (revised Figure S14b) of the mode effective indices (expressed by $n_{\text{effective}} + ik_{\text{effective}}$) of the ridge with non-vertical sidewalls as a function of the width at the wavelengths of 575, 580, and 585 nm. The corresponding $Q_{\text{leak-surr}}$ of the BSW ring cavities were then computed and reported in Figure S14c. The $Q_{\text{leak-surr}}$ of the BSW ring cavities exhibits a relatively weak wavelength dependency in the investigated wavelength range (the reported lasing wavelength range was ~ 10 nm, corresponding to a variation in $Q_{\text{leak-prop}}$ of ~ 10 %).

Figure S14. The analysis of the $Q_{\text{leak-surr}}$ of the BSW ring cavities at different wavelengths. The (a) real part and (b) imaginary part of the mode effective indices at wavelengths of 575 (black), 580 (red), and 585 nm (blue). (c) The simulated surrounding losses (defined as $Q_{\text{leak-surr}}$) of the BSW modes using mode analysis (diameter independent results). Although the $Q_{\text{leak-surr}}$ of the BSW ring cavities exhibits weak wavelength dependence (lower propagation loss for shorter wavelength due to higher optical confinement), a strong dependency with the ridge width is observed.

The revised parts of the manuscript are provided below: (Line 8 page 17) “*Note that the values of $Q_{\text{leak-surr}}$ ($\sim 10,000$ – $40,000$) are of the same order as the Q_{leak} of the BSW ring cavities with large diameters ($D = 50, 75, \text{ and } 100 \mu\text{m}$), suggesting that the light leakage to the surrounding is indeed a dominant loss for large ring cavities so that their Q_{leak} exhibit a weak ridge width dependence.*”

d) After looking at the field distribution in Fig. 5f, it seems that losses would be increased, because of the higher proximity of the BSW mode to the ridge interface. Instead, this approach provides an underestimation of losses. Why the eigenfrequency analysis misses a correct quantification of propagation losses? What is the point in using such a model?

After re-examing Q_{leak} and $Q_{\text{leak-surr}}$ (see definition and results in the previous reply), which are calculated according to the BSW laser structure geometry obtained from SEM images of the fabricated devices, we attribute the lasing threshold dependency on the ridge width mainly to the surface scattering losses because the $Q_{\text{leak-surr}}$ for the ridge widths of $w = 2$ and $2.5 \mu\text{m}$ shows only a weak dependency and the newly presented results (Fig. S15) on scattering losses show a clear dependency with the ridge width. We have deleted the statement about the underestimation of propagation losses in the eigenfrequency analysis and revised the **discussion part related to losses**. We have also added **Figure S15** to include simulated results about the dependency of the surface scattering losses with the ridge width.

The presented eigenfrequency analysis results should include the losses related to the lateral radiation leakage due to the effect of curvature (denoted as $Q_{\text{leak-curv}}$) and the leakage of light to the surrounding (denoted as $Q_{\text{leak-surr}}$), however, these results could not fully explain the dependency of the lasing threshold with the ridge width. We assumed in the first version of our manuscript that some losses (e.g., light leakage to the surroundings and surface scattering loss) should be responsible for this effect. However, the revised $Q_{\text{leak-surr}}$ (surrounding losses) shows only a weak dependency with the width, which contradicts the lasing threshold's strong dependency with the ridge width. Here, we show by simulation that surface scattering loss should be the main reason for the lasing threshold dependency with the ridge width. We have added

the effect of roughness (surface scattering) on the ridge in the mode analysis (roughness could not be added in the eigenfrequency analysis). The roughness effect is shown in Figure S15 and exhibits a clear dependency on the ridge width. Figures S15a–c show the calculated electric field norm distributions of the BSW mode ($d = 136$ nm) for the ridges $w = 1.5, 2,$ and 2.5 μm with smooth surfaces. In these figures, the electric field distributions of the guided BSW mode within the ridge become less confined with decreasing the ridge width, implying that more light can interact with the edge and therefore with surface roughness. To verify this hypothesis, we have introduced roughness on the surface of the ridge and performed the mode analysis to investigate the surface scattering losses. The calculated electric field norm distributions of the BSW modes ($d = 136$ nm) for the ridges ($w = 1.5, 2, 2.5$ μm) with surface roughness are shown in Figures S15d–f. The imaginary part ($k_{\text{effective}}$) of the mode effective indices is found to increase when the surface roughness is introduced, and the increase in $k_{\text{effective}}$ is largest for the smallest ridge of $w = 1.5$ μm . These simulation results confirm that scattering losses become larger with decreasing width of the ridge. From the above discussion, it becomes clear that scattering losses of the ring cavities should be another possible reason for the strong dependency of the lasing threshold on the ridge width.

We have added Figure S15 to include information about the dependency of the scattering losses with the ridge width.

Figure S15. Effect of surface roughness on the mode effective index for different ridge widths. (a–c) The calculated electric field norm distributions of the BSW modes ($d = 136$ nm) for (a) $w = 1.5$ μm , (b) $w = 2$ μm , and (c) $w = 2.5$ μm . (d–f) The calculated electric field norm distributions of the BSW modes ($d = 136$ nm) for (d) $w = 1.5$ μm , (e) $w = 2$ μm , and (f) $w = 2.5$ μm when surface roughness was introduced (root-mean-square roughness was set to 10 nm, a value corresponding to a tradeoff between computing time and mesh size). The real and imaginary parts of the mode effective indices (expressed by $n_{\text{effective}} + ik_{\text{effective}}$) are indicated in the figure. The simulated increase in $k_{\text{effective}}$ due to roughness are 5.1×10^{-5} , 4.8×10^{-5} , and 3×10^{-6} for the ridge widths w of 1.5, 2, and 2.5 μm , respectively. The increased losses are attributed to scattering, and the scattering losses become larger with decreasing width of the ridge.

Reviewer #3 (Remarks to the Author):

The manuscript reports on the development of a microlaser based on Bloch surface waves (BSW) by fabrication of polymer waveguides in the form of ring resonators on the surface of a one-dimensional photonic crystal (PC). Lasing manifests itself in the spectral region of Rhodamine 6G gain as a set of peaks typical of whispering gallery modes (WGM) and is observed at a subdiffractive waveguide thickness. As reference samples, the same ring resonators on a quartz substrate are used, for which the absence of lasing is shown at waveguide thicknesses below the diffraction limit. The possibility of sensing with fabricated lasers is shown.

The all-dielectric BSW platform is very flexible in designing nanophotonic devices for sensing and controlling light, operating in arbitrary spectral ranges. An additional attraction is the possibility of using low-index materials such as polymers for BSW control. In this regard, the development of microlasers for the BSW platform, which can be simply printed from polymers using industrial lithography techniques, is a significant and noteworthy task. However, the manuscript contains a number of inaccuracies and unclear results that should be clarified before the manuscript can be published. The following shortcomings can be distinguished:

1. The main claim that lasing is based on BSW propagating mode is not rigorously proven.

1.1. In proving the presence of BSW lasing, the authors mainly relied on calculations. However, the mode analysis of waveguides looks incomplete, and the results are presented carelessly.

What type of modes (TM or TE) are studied for each of the three systems?

In waveguides with a width of 1.5 μm or more, higher-order modes can exist (for instance, TE₀₁, TE₀₂, etc in BSW waveguides, see [1,2]). However, the manuscript deals only with fundamental modes.

Also, the possible excitation of volume waveguide modes of PC [3] or hybrid BSW-waveguide modes under the polymer ridge is not discussed. For example, can the polymer layer together with the top SiO₂ layer of PC be considered as a waveguide core in which the usual photonic mode can be excited (in this case, PC acts as a mirror, see for example [4])?

We would like first to thank the Reviewer for these important comments and provide a better description of our results in the following.

1. We have indicated in the manuscript that the photonic mode is a TE mode, the BSW mode is a TE-like guided mode, and the hybrid plasmonic mode is a TM mode.

Accordingly, some sentences have been revised as below. (Line 3 page 8) “*Mode analysis (see details in the Methods section) is first employed to investigate and characterize the optical characteristics of the BSW mode (transverse electric-like guided mode) guided by a R6G-doped SU-8 polymer ($n = 1.6$) ridge waveguide having a thickness of 140 nm on a dielectric multilayer. For comparison purposes, a photonic mode (transverse electric mode, guided by a 500- or 200-nm-thick polymer*

ridge waveguide on quartz) and a hybrid plasmonic mode (transverse magnetic mode, guided by a 140-nm-thick polymer ridge waveguide on an Ag substrate with a 10-nm SiO₂ interlayer) are also examined.”

2. To investigate the complete BSW modes including the higher order modes, we have comprehensively investigated the modes sustained by the proposed BSW laser structure. We have added a **discussion section “The detailed discussions about the verification of the BSW lasing”** and **Figure S11** in the Supporting Information.

In the following simulation results, the BSW laser structure geometry was obtained from SEM images of the fabricated devices, which comprise five pairs of alternating TiO₂ and SiO₂ layers having respective thicknesses of 82 and 170 nm (as the BSW platform) and an R6G-doped SU-8 ridge having a thickness of 136 nm (as the top guiding structure with the oblique sidewall) lying onto the surface of the BSW platform.

Since the width of the ridge waveguide is greater than or equal to 1.5 μm , the higher-order BSW modes indeed exist in the structure and they are investigated using mode analysis. Figures S11a–c show the calculated real parts (Figure S11a) and imaginary parts (Figure S11b) of the mode effective indices (expressed by $n_{\text{effective}} + ik_{\text{effective}}$) and the confinement factor (Figure S11c) of the guided zero-order, first-order, and second-order BSW modes within the R6G-doped SU-8 ridge as a function of the width w for a thickness d of 140 nm. Both the $n_{\text{effective}}$ and the confinement factor decrease while the $k_{\text{effective}}$ increases with increasing order of the mode for the BSW mode (for $w = 1.5 \mu\text{m}$, the $k_{\text{effective}}$ of the fundamental mode increases by 5 times and 30 times for the first- and second-order modes, respectively), implying that the higher-order BSW modes possess less optical confinement and higher propagation

losses. Therefore, the higher-order BSW modes are too leaky to be sustained as the lasing mode, while the fundamental BSW mode should be the natural mode for lasing in the BSW lasers.

Figure S11. Optical characteristics of the higher-order BSW modes as a function of the SU-8 polymer ridge width w with non-vertical sidewalls ($d = 136 \text{ nm}$). (a, b) The (a) real parts and (b) imaginary parts of the mode effective indices (expressed by $n_{\text{effective}} + ik_{\text{effective}}$). (c) The confinement factor. (d–f) Calculated electric field norm distributions of the (d) zero-order, (e) first-order, and (f) second-order BSW mode for $w = 1.5 \mu\text{m}$.

3. The internal modes of the dielectric multilayer (PC volume waveguide mode) and the possibility of conventional photonic modes are also discussed below, and we have added **Figure S12** in the Supporting Information.

The internal modes are Bloch waves which exist within a periodic multilayer structure¹. The characteristics of these modes are that their maximum electric field

distributions are mainly located in the center of the multilayer (Figure S12), implying the electric energy of such mode is confined at locations far away from the top fluorescent material (R6G-doped SU-8 polymer ridge). Thus, the coupling to internal modes is weak in the BSW laser structure. Moreover, the overlap between the electric field distribution of the internal modes and the R6G-doped SU-8 polymer (as the gain medium in this study) is also very small. Therefore, the optical feedback for such a mode is inadequate so the lasing action based on the stimulated emission of an internal mode should not be possible in BSW lasers.

Figure S12. The calculated one-dimensional electric field distributions for one of the representative internal modes of the BSW platform with a SU-8 polymer thickness of $d = 136$ nm.

Furthermore, using mode analysis, we can only find modes that are mainly confined in the polymer ridge (Figure R1). A photonic mode with an electric field uniformly distributed within the polymer ridge and the top SiO₂ layer (considering the polymer ridge and the top SiO₂ layer as a waveguide core) could not be found. This behavior can be attributed to the presence of the higher refractive index of the underneath TiO₂ layer which makes it unlikely for the polymer ridge and the top SiO₂ layer to form a waveguide core that supports a photonic mode.

Figure R1. The calculated electric field norm distributions of the BSW modes sustained within a SU-8 polymer ridge waveguide ($w = 1.5 \mu\text{m}$, with vertical sidewall) with thicknesses of (a) 100 nm, (b) 200 nm, and (c) 500 nm onto the surface of TiO₂/SiO₂ multilayer having respective thicknesses of 82 nm/170 nm.

The revisions of the manuscript are provided below: (Line 2 page 14) “*To confirm BSW lasing, discussions are provided about the higher modes in Figure S11 and the internal modes in Figure S12, and experimental evidence is also provided by leakage radiation microscopy in Figure S13.*” (for an explanation regarding the leakage radiation microscopy, see the replies below)

Reference

1. Badugu, R., Mao, J., Zhang, D., Descrovi, E., Lakowicz, J. R. Fluorophore Coupling to Internal Modes of Bragg Gratings. *J. Phys. Chem. C* **124**, 22743 (2020).

1.2. The authors experimentally determined the group index for the BSW ring resonators under study, but did not comment on the obtained value. Nevertheless, the group index allows us to estimate the effective refractive index of the supposedly excited mode if we know its dispersion (for example, from calculations). Considering the small dispersion of the materials used, one can assume that a PC volume waveguide mode with an effective refractive index close to the effective index (1.82) is excited in the resonator (such modes are usually easily observed inside a PC in calculations). An appropriate analysis should be performed to refute (or confirm) this assumption. It also makes sense to use the found group index for WGM photonic resonators in the analysis.

We have added a discussion section “*The discussion of the group index of the BSW lasers*” and **Figure S9** in the Supporting Information. The manuscript has been revised accordingly.

We theoretically calculated the group index n_g of the BSW lasers from the BSW dispersion diagram (Figure S9a) and compared its value to the value determined experimentally. Here, the BSW dispersion was obtained using the mode analysis. The group index is defined as the ratio of the velocity of light in vacuum (c) to the group velocity (v_g) for the BSW mode. The group velocity v_g is expressed by:

$$v_g = \frac{d\omega}{dk}, \text{ (S2)}$$

which can be directly obtained from the slope of the BSW dispersion. Accordingly, the group index n_g of the BSW laser is calculated to be 1.83 from the definition $n_g = c/v_g$, and this value agrees well with the experimentally determined group index of 1.82.

For comparison purposes, the group index n_g of the photonic lasers is also theoretically and experimentally investigated. First, the dispersion diagram of the SU-8 polymer ridge on a quartz substrate is calculated using the mode analysis, as displayed in Figure S9a. According to the definition, the group index n_g of the photonic lasers is estimated to be 1.72. On the other hand, to determine the group index n_g experimentally, photonic lasers ($d = 520$ nm) with different diameters ranging from 30 to 100 μm were optically pumped, and their lasing mode spacing $\Delta\lambda$ versus the inverse of the cavity dimension $1/(\pi D)$ is plotted in Figure S9c. From the linear dependence between $\Delta\lambda$ and $1/(\pi D)$, the group index n_g of the photonic lasers is estimated to be 1.70 using the theoretical prediction $\Delta\lambda = \lambda^2/(\pi D n_g)$. Note that this value agrees well with the theoretical value, and is different from that of the BSW lasers (Figure S9b). According to the investigation of the group index, lasing from the BSW lasers shows distinguishable characteristics from the photonic lasers, confirming that the lasing action of the BSW lasers is based on the stimulated emission of a BSW mode.

As discussed in the previous section, the internal mode (PC volume waveguide mode) of a periodic multilayer cannot be the lasing mode because the coupling to the internal mode and the optical feedback for such a mode are unfavorable. Instead, the gain mechanism of the BSW lasers should be based on the stimulated emission of a BSW mode.

Figure S9. The investigation of group index. (a) The BSW dispersion diagram of the $\text{TiO}_2/\text{SiO}_2$ multilayer having respective thicknesses of 82 and 170 nm with a 136-nm-thick top SU-8 polymer ridge (with non-vertical sidewall, blue). The dispersion diagram of the SU-8 polymer ridge on a quartz substrate (having a thickness of 520 nm and non-vertical sidewall in green) is also displayed. (b, c) Mode spacing analysis of the (b) BSW lasers and the (c) photonic lasers ($d = 520$ nm) as a function of their cavity dimensions. The mode spacing is inversely proportional to the diameter of the ring cavity.

The revised parts of the manuscript are provided below: (Line 2 page 13) “*This value agrees well with the calculated n_g obtained from the dispersion of the BSW mode and is significantly different from those of a photonic mode (see Figure S9 in the Supporting Information). Thus, these experimental results on the group index confirm the BSW lasing behavior of the proposed BSW lasers.*”

1.3. The BSW platform provides a unique opportunity to study the properties of BSW modes by analyzing the leakage radiation. Since the effective refractive index of BSW modes is lower than the refractive index of PC materials, in the case of transparent substrate, the leakage radiation can be collected using conventional immersion objective lenses [2, 5, 6] or prisms in the Kretschmann scheme [7,8]. This makes it possible not only to determine the mode composition of the BSW waveguides and effective refractive indices of modes, but also to directly visualize the BSW modes [2], which would help to clearly demonstrate the operation of the BSW laser, thereby significantly improving the manuscript.

Indeed the characterization by leakage radiation microscopy should be an important missing piece of evidence to our work. We have investigated the optical characteristic of the lasing mode of the BSW lasers by leakage radiation microscopy¹ and added the results in the section “*Investigation into the BSW lasing mode by leakage radiation microscopy*” and **Figure S13** in the Supporting Information.

The sample was firstly prepared by depositing a dielectric multilayer (BSW platform) comprising five TiO₂/SiO₂ pairs having respective thicknesses of 82 nm and 170 nm on a 100- μ m-thick quartz substrate. Then, R6G-doped SU-8 polymer ring cavities were fabricated onto the surface of the BSW platform through the top-down photolithography described in the Methods section. When the sample was optically pumped by the ns pulsed laser using the conventional microscopy configuration (Figure S7), the leakage radiation from the sample was collected simultaneously with an oil immersion objective lens having a numerical aperture (NA) of 1.4. A band-pass filter with a central wavelength of 580 nm and a full width at half maximum (FWHM) of 10 nm was placed in the optical path after the oil immersion objective lens to filter out the leakage radiation light from the stimulated process only. Then, the leakage

radiation is transmitted through a tube lens and split by a 50:50 beam splitter. On one side of the beam splitter, the leakage radiation was captured by a CCD camera for optical image observation. On the other side, a convex lens was used to collect the radiation light to produce a back focal plane (BFP) image on a CCD camera (CS505MU, Thorlabs, Inc.), which is the Fourier transform of the direct plane image. By analyzing the BFP image of the BSW lasers, the wave-vector information (e.g., the effective refractive index) of the lasing mode can then be obtained.

Figure S13 shows the BFP images of the BSW laser having a diameter D of 100 μm ($d = 136$ nm and $w = 2.5$ μm) obtained below (Figure S13a) and above (Figure S13b) the lasing threshold. Several rings are observed, and each ring corresponds to a leakage radiation mode from the BSW laser. The effective refractive index ($n_{\text{effective}}$) of the mode can be expressed by NA in the BFP image and is derived by the relation:

$$n_{\text{effective}} = \text{NA in the BFP image} = 1.4 \times \frac{R_{\text{mode}}}{R_{\text{edge}}}, \quad (\text{S3})$$

where R_{mode} is the radius of the ring, R_{edge} the radius of the edge of the BFP image, which corresponds to the NA of the oil immersion objective lens used in the setup (NA = 1.4). In Figure S13a, the inner ring (NA = 1.06) corresponds to the BSW mode sustained by the bare multilayer (without top R6G-doped SU-8 polymer ridge), and the outer rings should correspond to multilayer modes, including the internal modes and the BSW-guided modes within the SU-8 ridge waveguide. When the pump energy density increases above the lasing threshold of the BSW laser (Figure S13b), all of the rings in the BFP image become brighter, and a very bright ring is observed with NA \sim 1.32 to 1.4, which can be attributed to the leakage radiation of the lasing emissions. Note that the NA value of the lasing emission agrees well with the calculated $n_{\text{effective}}$ of the BSW mode ($n_{\text{effective}} = 1.36$, as seen in Figure S15c).

According to these results, the lasing mode of the BSW lasers is confirmed to be indeed the BSW mode.

Figure S13. BFP images of the BSW laser having a diameter D of $100\ \mu\text{m}$ ($d = 136\ \text{nm}$ and $w = 2.5\ \mu\text{m}$) obtained (a) below and (b) above the lasing threshold using leakage radiation microscopy.

Reference

1. Descrovi, E. *et al.* Leakage radiation interference microscopy. *Opt. Lett.* **38**, 17, 3374–3376 (2013).

2. The applicability of developed BSW lasers is discussed too briefly.

2.1. The authors state that the BSW lasers possess a low lasing threshold. Could the authors compare this value with lasing thresholds of others on-chip WGM lasers?

As the material properties of the gain media employed in the lasing structures to demonstrate on-chip lasers play an important role in the lasing threshold values, we have summarized the lasing performances of on-chip WGM lasers made of R6G-doped polymer lasers in Table S2.

The lasing threshold of our fabricated BSW laser ($6.7 \mu\text{J}/\text{mm}^2$) is found to be comparable to those of the reported photonic WGM lasers (a few $\mu\text{J}/\text{mm}^2$). We also note that the thickness of the guiding structure of the proposed BSW laser is much smaller than those of the reported on-chip WGM lasers based on photonic modes. In summary, we have added Table S2 in the Supporting Information and deleted the statement of “low lasing threshold” for our measured value of the lasing threshold.

Table S2. On-chip WGM lasers based on dye-doped polymers.

Gain medium (method)	Waveguide mode	Guiding structure thickness	Threshold	Pulse duration of pump laser	Year	Ref.
BEH-PPV	Photonic mode	NA	$10 \mu\text{J}/\text{mm}^2$	10 ns	1998	[1]
R6G doped SU-8	Photonic mode	48 μm	$25 \mu\text{J}/\text{mm}^2$	5 ns	2005	[2]
R6G doped PMMA	Photonic mode	1 μm	$2.5 \mu\text{J}/\text{mm}^2$	15 ns	2010	[3]
PM597 doped resist	Photonic mode	1 μm	$12 \mu\text{J}/\text{mm}^2$	5 ns	2011	[4]
PM597 doped PMMA	Photonic mode	$\sim 1 \mu\text{m}$	$0.3 \mu\text{J}/\text{mm}^2$	20 ns	2015	[5]
R6G doped SU-8	Photonic mode	12 μm	$0.2 \mu\text{J}/\text{mm}^2$	10 ns	2015	[6]
R6G-doped TZ-001	Photonic mode	30 μm	$9.3 \mu\text{J}/\text{mm}^2$	5 ns	2017	[7]
R6G doped SU-8	Photonic mode	30 μm	$2.5 \mu\text{J}/\text{mm}^2$	5 ns	2018	[8]
R6G doped SU-8	Photonic mode	0.24 μm (cutoff of photonic mode)	$100 \mu\text{J}/\text{mm}^2$	0.5 ns	2023	This work
R6G doped SU-8	BSW mode	0.14 μm	$6.7 \mu\text{J} / \text{mm}^2$	0.5 ns	2023	This work

References

1. Kawabe, Y. *et al.* Whispering-gallery-mode microring laser using a conjugated polymer. *Appl. Phys. Lett.* **72**, 141 (1998).
2. Nilsson, D., Balslev, D., Gregersen, M. M., Kristensen, A. Microfabricated solid-state dye lasers based on a photodefinable polymer. *Appl. Opt.* **44**, 23, 4965–4971 (2005).
3. Grossmann, T. *et al.* Low-threshold conical microcavity dye lasers. *Appl. Phys. Lett.* **97**, 063304 (2010).
4. Grossmann, T. *et al.* Direct laser writing for active and passive high-Q polymer microdisks on silicon. *Opt. Express* **19**, 12, 11451–11456 (2011).
5. Wienhold, T. *et al.* All-polymer photonic sensing platform based on whispering-gallery mode microgoblet lasers. *Lab Chip* **15**, 3800–3806 (2015).
6. Chandralalim, H., Fan, X. Reconfigurable Solid-state Dye-doped Polymer Ring Resonator Lasers. *Sci. Rep.* **5**, 18310 (2015).
7. Wan, L. *et al.* On-chip, high-sensitivity temperature sensors based on dye-doped solid-state polymer microring lasers. *Appl. Phys. Lett.* **111**, 061109 (2017).
8. Wan, L. *et al.* Demonstration of versatile whispering-gallery micro-lasers for remote refractive index sensing. *Opt. Express* **26**, 5, 5800–5809 (2018).

2.2. In Ref. 26, the efficiency of coupling laser radiation into the BSW modes is estimated to be at least 15%, while the lasing threshold is 20 times lower at a smaller laser size.

Could the authors estimate the efficiency of coupling radiation from their ring BSW laser into BSW modes, for example, of a straight waveguide and compare it with the case from Ref. 26? Demonstrating the possibility of using the BSW laser for integrated photonics, at least through simulation, would improve the manuscript.

We have added a section discussing the coupling between a BSW ring laser cavity and a waveguide entitled “*Coupling between a BSW ring laser cavity and a waveguide*” and **Figure S19** in the Supporting Information.

To demonstrate the possibility of using the BSW laser for optical integrated circuits, we have investigated the coupling between a BSW ring laser cavity and a waveguide using the finite-difference time-domain (FDTD) technique (COMSOL software). Due to the large diameter of the ring cavity ($D = 30\text{--}100\ \mu\text{m}$), the simulation model can be simplified as two parallel straight ridge waveguides having a gap between them (Figures S19a, b). A BSW mode is launched from one end of the input ridge waveguide and propagates along the y -direction. Some portion of the guiding BSW mode will couple to the adjacent waveguide and then propagates to the end of the output ridge waveguide. Then, the coupling efficiency is estimated as the ratio between the power at the output ridge and the input ridge ($P_{\text{out}}/P_{\text{in}}$). Figure S19c shows the simulated coupling efficiency for the BSW laser as a function of the gap between the two SU-8 polymer ridge waveguides. The coupling efficiency increases with decreasing the gap, and a value of approximately 2% is obtained for a gap of

0.050 μm . Note that the estimated coupling efficiency is underestimated using the current model because it does not take into account the effect of the curvature of the ring laser cavity. Thus, we believe that an efficient coupling between a BSW ring laser and a waveguide can be obtained. Accordingly, the applicability of the proposed BSW lasers in integrated optical circuits is established so that future applications of BSW lasers on a chip such as optical modulators and logic devices can be realized.

Regarding the comparison with the coupling efficiency of laser radiation into a BSW mode reported in Ref 26 (Ref 34 in the revised manuscript), we should like to point out that this reference reports the directional coupling of BSW mode on the BSW platform without a guiding structure and is therefore different from our work where the coupling of a lasing mode into an on-chip waveguide is described.

Figure S19. The investigation of coupling between a BSW ring laser cavity and a

waveguide. (a, b) Schematic representation of the (a) coupling between a BSW ring laser cavity and a straight BSW waveguide, and (b) top view (left) and cross-sectional view (right) of the simulation model. The width of the SU-8 polymer ridge is 1.5 μm . (c) The estimated coupling efficiency between the BSW ring laser cavity and the waveguide as a function of the gap.

The revised part of the manuscript is provided below: (Line 15 page 18) “*In the following, we discuss possible applications of the proposed BSW lasers by first showing the possibility for on-chip integration and then reporting a sensitivity investigation. The coupling between a BSW ring laser cavity and a waveguide, which is required for on-chip integration, is investigated by simulation in Figure S19.*”

2.3. How does the sensing performance of the BSW laser compare with other WGM lasers used in sensing [9]?

We have added **Figure S21c** in the Supporting Information to compare the sensitivity between the cases of the photonic laser and BSW laser.

In this figure, we compare the sensitivity of the photonic laser made of R6G-doped SU-8 polymer having a thickness of $d = 520$ nm with that of the BSW laser ($d = 136$ nm) for the diameter ($D = 100$ μm). The wavelength shifts are computed as a function of the increase in the ridge size Δa caused by swelling of the polymer using eigenfrequency analysis for axisymmetric models. When the ridge size increases, the resonant wavelength of the photonic laser red-shifts linearly, but the amount of the wavelength shifts is just 12% compared to that of the BSW laser. This result suggests a high sensitivity for the BSW lasers.

Further comparison with reported photonic WGM-based laser devices is made difficult due to the different types of targeted analytes (biomaterials, chemicals in the gas phase, etc) and sensing mechanisms (shifts caused by a change in surrounding bulk refractive index, surface modification).

Figure S21. The simulated wavelength shifts of the BSW ring lasers with respect to (a) the increase in ridge size Δa and (b) the increase in refractive index Δn . (c) The simulated wavelength shifts of the photonic ring lasers ($d = 520$ nm) with respect to the increase in ridge size Δa . The ring diameter for both BSW laser and photonic laser is $100 \mu\text{m}$.

The revisions of the text in the Supporting Information are provided below:

(Line 10 page 26) “*For comparison purposes, the simulated wavelength shifts of the photonic laser ($d = 520$ nm) with respect to the increase in the ridge size (with constant refractive index) are also displayed in Figure S21c.*”

(Line 16 page 26) “*Note that when the ridge size increases, the resonant wavelength of the photonic laser red-shifts linearly, but the amount of the wavelength shifts is just 12 % compared to that of the BSW laser. This result suggests a high sensitivity for the BSW lasers.*”

3. Some of the results are presented carelessly. Not all methods are described in sufficient detail to reproduce the work.

3.1. Lines 136-137: «Mode analysis (see details in the Methods section) is first employed to design ... BSW mode». I didn't find in the manuscript how mode analysis is applied to design BSW laser. Moreover, the PC design method is also not fully outlined in Supporting Information.

In particular, what angle theta was taken? Why does the PC consist of only 5 pairs of layers? Increasing the number of layers should reduce radiation losses [10], which can improve the lasing properties. In general, the optimization of radiation losses in BSW waveguides is not a trivial task [11, 12].

We have revised our manuscript to give a detailed description of the employed methods.

(Line 3 page 8) *“Mode analysis (see details in the Methods section) is first employed to investigate and characterize the optical characteristics of the BSW mode guided by an R6G-doped SU-8 polymer ($n = 1.6$) ridge waveguide having a thickness of 140 nm on a dielectric multilayer.”*

(Line 8 page 5 in the Supporting Information) *“According to this equation, the thicknesses of TiO_2 and SiO_2 layers are 71 and 154 nm, respectively, which are designed at a refraction angle θ of 50° within the SiO_2 layer (corresponding to a refraction angle within the TiO_2 layer of 36.1°).”*

Regarding the design of the multilayer, increasing the number of pairs will reduce the propagation losses (for example, the $k_{\text{effective}}$ of the BSW modes for the SU-8 polymer ridge having a thickness $d = 140$ nm and a width $w = 2.5$ μm are 1.8×10^{-5} ,

1.6×10^{-5} , and 1.2×10^{-5} for the pair numbers of 5, 7, and 10, respectively) and thus improve the lasing performance of the proposed BSW laser. However, fabricating a multilayer with a larger number of pairs will require a longer deposition time so a tradeoff between the improvement of the lasing performance and the fabrication time should be found.

3.2. Lines 262-263: «Due to the transparent nature of the SU-8 polymer, the Q_{abs} of the ring cavities is of the order of 10^8 [Ref. 45]». In Ref. 45 there is nothing about SU-8, only about PMMA. Moreover, as follows from Figure S1, the absorption of R6G/ethanol/SU-8 solution is not zero inside the lasing band.

Thank you for pointing out this mistake. We have revised the reference related to the optical constants of the SU-8 polymer (MicroChem. SU-8 2000 Permanent epoxy negative photoresist: Processing guidelines for SU-8 2000.5, SU-8 2002, SU-8 2005, SU-8 2007, SU-8 2010 and SU-8 2015. Available from:

<http://www.microchem.com/pdf/SU-82000DataSheet2025thru2075Ver4.pdf> (2017).

We have also revised the manuscript as below:

(Line 20 page 14) “*Due to the transparent nature of the SU-8 polymer, the absorption loss ($1/Q_{abs}$) of the ring cavities should be low so that it can be neglected in the analysis of the loss mechanisms⁵³.*”

We noticed that the absorption of the R6G/ethanol/SU-8 solution is not zero in the lasing band of the R6G-doped SU-8 ring lasers, which is attributed to the absorption of the R6G dyes. However, in our study where the lasing threshold of a laser device is investigated, the absorption of the gain medium is usually taken as zero because the gain medium should be transparent to the mode when pumped at energy density near the lasing threshold (at which the gain saturation occurs)¹⁻³. Accordingly, we believe that it should be reasonable to neglect the absorption of the gain medium when investigating the gain mechanisms of the BSW laser cavities.

References

1. Elbaz, A. *et al.* Reduced lasing thresholds in GeSn microdisk cavities with defect management of the optically active region. *ACS Photonics* **7**, 2713–2722 (2020).

2. Cho, S., Yang, Y., Soljačić, M., Yun, S. H. Submicrometer perovskite plasmonic lasers at room temperature. *Sci. Adv.* **7**, eabf3362 (2021).
3. Tang, S. J. *et al.* Laser particles with omnidirectional emission for cell tracking. *Light Sci. Appl.* **10**, 23 (2021).

3.3. How was eigenfrequency analysis performed, and which axisymmetric model was used?

We performed the eigenfrequency analysis using the COMSOL software. Eigenfrequency corresponds to a frequency at which a system is prone to experience a resonance in the absence of any external driving force. In eigenfrequency analysis, the corresponding eigenmode profiles—here, the whispering gallery mode—of a ring resonator are obtained, and the losses (the imaginary part of the eigenfrequencies) and the quality factors ($Q = \omega_{\text{real}}/2\omega_{\text{imag}}$, where ω_{real} and ω_{imag} are the real and imaginary parts of the calculated eigenfrequency, respectively) of the mode can be calculated. According to this characteristic, eigenfrequency analysis is generally applied to study the resonant properties as a function of a physical parameter.

Before calculating the eigenfrequency of the resonant system, a simulation model describing the geometric configuration and the optical properties should be established. Since our proposed ring cavity possesses a rotation symmetry, we have used a two-dimensional (2D) cross-section of the cavity (including the ridge waveguide and the underneath multilayer sustaining the BSW mode) and incorporated the axisymmetry to form a quasi-3D model for calculation. This simulation method has been validated numerically, analytically, and experimentally in a past study¹, and such an axisymmetry model reduces the calculation time and memory so that the eigenfrequency analysis can be performed more efficiently.

Reference

1. Cheema, M. I., Kirk, A. G. Accurate determination of the quality factor and tunneling distance of axisymmetric resonators for biosensing applications. *Opt. Express* **21**, 7, 8724–8735 (2013).

3.4. Lines 316-318: «... implying that more light can interact with the edge or surface roughness (see Figure S9b–d in the Supporting Information)». Such a statement can hardly be made on the basis of Figure S9b–d.

We have added **Figure S15** to include information about the dependency of the scattering losses with the ridge width.

We have added the effect of roughness (surface scattering) on the ridge in the mode analysis (roughness could not be added in the eigenfrequency analysis). The roughness effect is shown in Figure S15 and exhibits a clear dependency on the ridge width. Figures S15a–c show the calculated electric field norm distributions of the BSW mode ($d = 136$ nm) for the ridges $w = 1.5, 2,$ and 2.5 μm with smooth surfaces. In these figures, the electric field distributions of the guided BSW mode within the ridge become less confined with decreasing the ridge width, implying that more light can interact with the edge and therefore with surface roughness. To verify this hypothesis, we have introduced roughness on the surface of the ridge and performed the mode analysis to investigate the surface scattering losses. The calculated electric field norm distributions of the BSW modes ($d = 136$ nm) for the ridges ($w = 1.5, 2,$ 2.5 μm) with surface roughness are shown in Figures S15d–f. The imaginary part ($k_{\text{effective}}$) of the mode effective indices is found to increase when the surface roughness is introduced, and the increase in $k_{\text{effective}}$ is largest for the smallest ridge of $w = 1.5$ μm . These simulation results confirm that scattering losses become larger with decreasing width of the ridge. From the above discussion, it becomes clear that scattering losses of the ring cavities should be the possible reason for the strong dependency of the lasing threshold on the ridge width.

Figure S15. Effect of surface roughness on the mode effective index for different ridge widths. (a–c) The calculated electric field norm distributions of the BSW modes ($d = 136$ nm) for (a) $w = 1.5$ μm , (b) $w = 2$ μm , and (c) $w = 2.5$ μm . (d–f) The calculated electric field norm distributions of the BSW modes ($d = 136$ nm) for (d) $w = 1.5$ μm , (e) $w = 2$ μm , and (f) $w = 2.5$ μm when the surface roughness was introduced (root-mean-square roughness was set to 10 nm, a value corresponding to a tradeoff between computing time and mesh size). The real and imaginary parts of the mode effective indices (expressed by $n_{\text{effective}} + ik_{\text{effective}}$) are indicated in the figure. The simulated increase in $k_{\text{effective}}$ due to roughness are 5.1×10^{-5} , 4.8×10^{-5} , and 3×10^{-6} for the ridge widths w of 1.5, 2, and 2.5 μm , respectively. The increased losses are attributed to scattering, and the scattering losses become larger with decreasing width of the ridge.

3.5. Lines 439-440: «To investigate the gain mechanism of the proposed BSW lasers, the radiation-limited quality factor Q_{rad} was calculated...». What is the the radiation-limited quality factor Q_{rad} ?

We have revised the sentence in line 18 page 24, “To investigate the gain mechanism of the proposed BSW lasers, the quality factor (Q_{leak}) related to the lateral radiation leakage due to the effect of curvature and the leakage of light to the air and/or the substrate during the light propagation was calculated using the results of the eigenfrequency analysis performed on an axisymmetric model implemented in the COMSOL software.”

3.6. The pump energy density scale is confusing in Fig. 3b.

We have corrected the pump energy density scale in **Figure 3b**.

Figure 3. Lasing performance of BSW lasers. (a) Emission spectra under different pump energy densities and (b) output emission intensity as a function of pump energy density for a ring cavity having $d = 136 \text{ nm}$, $w = 1.5 \mu\text{m}$, and $D = 100 \mu\text{m}$. (c) The magnified emission spectrum of a BSW laser having $d = 136 \text{ nm}$, $w = 1.5 \mu\text{m}$, and $D = 50 \mu\text{m}$.

- [1] Sfez, T. et al. Bloch surface waves in ultrathin waveguides: near-field investigation of mode polarization and propagation. *J. Opt. Soc. Am. B* 27, 1617 (2010).
- [2] Safronov, K. R. et al. Multimode Interference of Bloch Surface Electromagnetic Waves. *ACS Nano* 14, 10428–10437 (2020).
- [3] Yeh, P., Yariv, A., & Hong, C. S. Electromagnetic propagation in periodic stratified media. I. General theory. *JOSA*, 67(4), 423-438 (1977).
- [4] Liscidini, M. Surface guided modes in photonic crystal ridges: the good, the bad, and the ugly. *J. Opt. Soc. Am. B* 29, 2103 (2012).
- [5] Descrovi, E. et al. Leakage radiation interference microscopy. *Opt. Lett.* 38, 3374 (2013).
- [6] Safronov, K. R. et al. Miniature Otto Prism Coupler for Integrated Photonics. *Laser and Photon. Rev.* 16, 2100542 (2022).
- [7] Moskalenko, V.V. et al. Surface wave-induced enhancement of the Goos-Hänchen effect in one-dimensional photonic crystals. *Jetp Lett.* 91, 382–386 (2010).
- [8] Pidgayko, et al. Direct imaging of isofrequency contours of guided modes in extremely anisotropic all-dielectric metasurface. *ACS Photonics*, 6(2), 510-515 (2018).
- [9] Toropov, N., Cabello, G., Serrano, M.P. et al. Review of biosensing with whispering-gallery mode lasers. *Light Sci Appl* 10, 42 (2021).
- [10] Vosoughi Lahijani, B. et al. Centimeter-Scale Propagation of Optical Surface Waves at Visible Wavelengths. *Advanced Optical Materials*, 2102854 (2022).
- [11] Perani, T. & Liscidini, M. Long-range Bloch surface waves in photonic crystal ridges. *Opt. Lett.* 45, 6534 (2020).
- [12] Luo, H., Tang, X., Lu, Y., & Wang, P. Low-Loss Photonic Integrated Elements Based on Bound Bloch Surface Wave in the Continuum. *Physical Review Applied*, 16(1), 014064 (2021).

(The reference list from Reviewer 3 was very useful to revise our manuscript)

REVIEWER COMMENTS

Reviewer #2 (Remarks to the Author):

The authors have diligently responded to all of my comments. I am happy to recommend publication.

Reviewer #3 (Remarks to the Author):

After having considered the extensive changes the authors operated on the manuscript and the detailed rebuttal letter, I think all the criticisms raised by the reviewers have been exhaustively and properly addressed. I would like to express my appreciation to the authors for the big effort done in this direction. The manuscript is definitely improved now and, in my opinion, it deserves publication on Nature Communications in its present form.

Reviewer #4 (Remarks to the Author):

The authors have done a lot of work and have improved the manuscript in many parts. In particular, new experiments were carried out and laser generation via BSW modes was confirmed. However, some points still remain questionable, reducing the quality of presentation.

1. The proposed sensing mechanism related to the swelling effect of SU-8 polymer due to diffusing toluene into SU-8 is not confirmed in the manuscript. The authors refer to Ref. 56, which did not study the effect of toluene on SU-8 at all. Ref. 56 considered the diffusion of water and some solvents into SU-8 (not all of them penetrated the polymer). Moreover, the polymer films were not subjected to a hard bake process, which could improve the chemical stability of SU-8, according to the authors. In the manuscript, BSW resonators were hard baked, further confirming that Ref. 56 cannot be used to describe the sensing mechanism. The authors should prove the diffusion of toluene into SU-8 from vapour through more relevant references. It is also worth noting that in almost all works on BSW sensors [1, 2], including gas sensors [3, 4], the mechanism of their operation is associated with a change in the refractive index of the adjacent medium or with the deposition of molecules on the surface of the BSW platform. Could the authors discuss this in the manuscript?

2. The numerical analysis of the losses in BSW resonators does not look self-consistent. First, as indicated in the main text, Q-factors are calculated using eq. (2) and (3) for BSW ring cavities, implying that Q_{leak} must be exactly equal to $Q_{\text{leak-surr}}$. If the values in eq. (3) are calculated for straight waveguides, this should be clearly reflected in the main text of the manuscript. However, in this case, the statement «the light leakage to the surrounding is indeed a dominant loss» is incorrect. As an example, for $D = 100$ nm and waveguide width of 2.5 μm , $Q_{\text{leak}} = 20000$ (Fig. 5b) and $Q_{\text{leak-surr}} = 40000$ (Fig. S14c). Thus, $Q_{\text{leak-curv}} = 40000$, i.e., for the largest cavity, the losses due to curvature is equal to losses due to leakage in surrounding and will only increase with decreasing diameter. However, in the case of $D = 100$ nm and waveguide width of 1.5 μm , $Q_{\text{leak}} \approx Q_{\text{leak-surr}} \approx 15000$ meaning no losses due to curvature. The authors should discuss this discrepancy.

3. Technical comments.

3.1. The correctness of the citation should be carefully checked:

3.1.1. The WGM size dependence of the lasing wavelength is proved by Ref. 51, in which a very specific system of spherical microdroplet-lasers is studied. Could the authors refer to works where the size dependence effect was studied in WGM lasers on photonic

platforms and briefly explain this effect in the manuscript?

3.1.2. Incorrect reference 53 before the eq. (1).

3.1.3. There are no data on the pump fluence and the pump beam size in Ref. S21 in supporting information.

3.2. As the WGM lasing threshold strongly depends on the WGM diameter, the latter should be added in the Table S2.

3.3. As it turned out that experimental values of PC layer thicknesses differ from that used in initial calculations, it should be specified for which PC parameters the calculations were carried out in each case (especially for Fig. 5).

3.4. For what length of waveguides is the coupling efficiency calculated in Fig. S19? I do not believe that the coupling efficiency was underestimated. On the one hand, the coupling can be enhanced due to the shift of the waveguide mode toward the outer side of the curved waveguide. On the other hand, as the curved waveguide moves away from the straight one, the coupling efficiency decreases. The more rigorous calculations are needed.

3.5. According to Snell's law, the refraction angle θ inside TiO₂ should be approximately 29 degrees instead of 36.1 degrees.

[1] Konopsky, V. N. & Alieva, E. V. Photonic crystal surface waves for optical biosensors. *Anal. Chem.* 79, 4729–4735 (2007).

[2] Sinibaldi, A. et al. Direct comparison of the performance of Bloch surface wave and surface plasmon polariton sensors. *Sensors Actuators, B Chem.* 174, 292–298 (2012).

[3] Descrovi, E., "Coupling of surface waves in highly defined one-dimensional porous silicon photonic crystals for gas sensing applications", *Appl. Phys. Lett.* 91, 241109 (2007).

[4] Michelotti, F., et al. "Fast optical vapour sensing by Bloch surface waves on porous silicon membranes», *Physical Chemistry Chemical Physics*, 12(2), 502-506 (2010).

Short note to the attention of the Reviewers:

In our replies, we first provide a list of important revisions (include the five new Figures S16, S17, S23, S24a and S24b, and the revised Figure S21) followed by a list of minor revisions. Our point-by-point replies are found after these lists. Also the revised manuscript (without and with **highlights**) and the Supporting Information (without and with **highlights**) can be found in the submission system. We would like to thank the Reviewer for his/her important comments.

List of important revisions

Additional experimental data and discussions

1. We have investigated the interaction between the SU-8 polymer and toluene vapor using a quartz crystal microbalance (QCM) detection system to clarify the sensing mechanism of the proposed BSW lasers and added a discussion and **Figure S23** in the section “**Environmental sensitivity investigation of the BSW lasers**” in the Supporting Information. **These experimental results show that the sensing mechanism of the proposed BSW lasers should be related to the sorption of toluene vapor or sorption followed by the swelling effect of the SU-8 polymer ridge.** We have also added **Figures S24a and S24b** to include a sensitivity study of the lasing wavelength to the sorption of toluene vapor onto the surface of SU-8 polymer. The manuscript has been revised accordingly.
2. We have added a discussion section “**Detailed discussions about the Q_{leak} of the BSW ring cavities**” and **Figures S16 and S17** in the Supporting Information and revised the manuscript accordingly. We have also revised the manuscript to state

that we evaluated the light leakage to the surrounding via Equation 3 by considering a straight waveguide, which sustains the propagating BSW mode. **These simulation results show that the light leakage to the surrounding is indeed a dominant loss for the proposed BSW ring cavities.**

3. We have revised the simulation for the coupling between a BSW ring laser cavity and a BSW waveguide. The manuscript text in the section reporting the “**Coupling between a BSW ring laser cavity and a waveguide**” and **Figure S21** in the Supporting Information have been revised accordingly.

Additional revisions related to discussion sections and references

1. We have added one sentence in the manuscript to state that **we used the mode analysis to discuss the effect of the roughness on the losses and explain the ridge width dependence on the lasing threshold.**
2. We have added reference 52 to our manuscript related to **the size dependence of the lasing spectrum** and added some descriptions of the cited works in the manuscript.
3. We have revised the incorrect reference to our manuscript related to **the total quality factor of an optical cavity in terms of the different loss mechanisms**, which can be expressed by Equation 1 (Reference 54: Krämmer, S. *et al.* Size-optimized polymeric whispering gallery mode lasers with enhanced sensing performance. *Opt. Express* **25**, 7884–7894 (2017)).
4. We have added a footnote in Table S2 to indicate how we estimated the lasing threshold for the on-chip WGM laser of reference S21.

5. We have added the diameter of the WGM lasers in Table S2.
6. We have specified the thicknesses of TiO₂ and SiO₂ layers for each simulation in the figure captions of Figures 1 and 5 in the manuscript and Figures S3, S6, S11, S14, S15, and S24 in the Supporting Information.
7. We have revised the refraction angle θ of light within the TiO₂ layer to be 29° in the Supporting Information.

Point-by-point responses to Reviewers' comments

REVIEWER COMMENTS

Reviewer #2 (Remarks to the Author):

The authors have diligently responded to all of my comments. I am happy to recommend publication.

We would like to thank Reviewer 2 for recommending the publication of our manuscript.

Reviewer #3 (Remarks to the Author):

After having considered the extensive changes the authors operated on the manuscript and the detailed rebuttal letter, I think all the criticisms raised by the reviewers have been exhaustively and properly addressed. I would like to express my appreciation to the authors for the big effort done in this direction. The manuscript is definitely improved now and, in my opinion, it deserves publication on Nature Communications in its present form.

We would like to thank Reviewer 3 for his/her kind comments and for recommending the publication.

Reviewer #4 (Remarks to the Author):

The authors have done a lot of work and have improved the manuscript in many parts. In particular, new experiments were carried out and laser generation via BSW modes was confirmed. However, some points still remain questionable, reducing the quality of presentation.

1. The proposed sensing mechanism related to the swelling effect of SU-8 polymer due to diffusing toluene into SU-8 is not confirmed in the manuscript. The authors refer to Ref. 56, which did not study the effect of toluene on SU-8 at all. Ref. 56 considered the diffusion of water and some solvents into SU-8 (not all of them penetrated the polymer). Moreover, the polymer films were not subjected to a hard bake process, which could improve the chemical stability of SU-8, according to the authors. In the manuscript, BSW resonators were hard baked, further confirming that Ref. 56 cannot be used to describe the sensing mechanism. The authors should prove the diffusion of toluene into SU-8 from vapour through more relevant references. It is also worth noting that in almost all works on BSW sensors [1, 2], including gas sensors [3, 4], the mechanism of their operation is associated with a change in the refractive index of the adjacent medium or with the deposition of molecules on the surface of the BSW platform. Could the authors discuss this in the manuscript?

We have revised the section “*Environmental sensitivity investigation of the BSW lasers*” by adding a discussion and **Figure S23** to clarify the sensing mechanism of the proposed BSW lasers, and added **Figures S24a and S24b** to include the sensitivity study of the lasing wavelength to the sorption of toluene vapor onto the surface of SU-8 polymer. We have also added reference 58 where volatile organic compounds including toluene are detected via the swelling of a SU-8 optical disk

resonator. The manuscript was revised accordingly.

We agree that in general the mechanisms for sensors are mainly associated with the change in the bulk refractive index of the adjacent medium or with the deposition of molecules on the surface of the devices. Besides, for polymer-based sensor devices, the swelling of the polymer due to the diffusion of the gas molecules is also expected to play a role in the sensing mechanism of gas vapors. In our work, we investigated the environmental sensitivity of the proposed BSW lasers by monitoring the lasing behavior in the presence of toluene vapor. Although hard-baking is employed to improve the stability of the proposed BSW lasers, the swelling of the SU-8 polymer is still possible as reported in a previous study¹. Therefore, to clarify the sensing mechanism of the proposed BSW lasers, we investigated the interaction between the SU-8 polymer and toluene vapor using a quartz crystal microbalance (QCM)² detection system. The well-known detection mechanism of QCMs relies on the shift of the resonance frequency of a quartz crystal resonator (QCR) upon deposition of a thin film on one of the sides of the quartz crystal. Transversal oscillation modes (e.g., breathing modes) are used to reduce interaction with the atmosphere just above the quartz crystal and thus realize a high-quality factor of resonance that enables high sensitivity to the thickness of deposited layers onto the quartz crystal.

An R6G-doped SU-8 polymer layer was deposited onto the surface of a QCR with a nominal resonance frequency of 5 MHz (CRTM-1000, ULVAC). To ensure that the properties of the SU-8 polymer layer are nearly identical to those of the BSW lasers, the same fabrication process was employed as for the BSW lasers, except for the presence of the photomask. Also, a bare QCR without a SU-8 polymer layer was prepared for comparison purposes. The QCRs were then subjected to toluene vapor by inserting them into a beaker containing liquid drops of toluene (the QCRs are not

immersed in toluene and the beaker is sealed to reach saturated toluene vapor) and their resonance frequencies were monitored and converted to a change in thickness using the parameters for a carbon layer. Figure S23a displays the observed change in the thickness of the QCRs without and with the SU-8 polymer layer when exposed to toluene vapor. In the presence of toluene vapor, the estimated thickness of the SU-8 polymer-coated QCR increased to approximately 35 Å within 300 s. In contrast, the thickness on the bare QCM remained the same. This result confirms that the SU-8 polymer interacts with toluene vapor. This interaction shown in Figure S23a could be related to the condensation and sorption of toluene on the surface and/or diffusion of toluene molecules into the SU-8 polymer (swelling). Because a change in the thickness is not observed on the bare QCR (negligible condensation and sorption on the metal surface of the bare QCR) and the relatively large change in the estimated thickness of the SU-8 polymer recovers (complete recovery is not reached due to the limited testing time) when the SU-8 polymer-coated QCR is removed from the toluene vapor, we infer that both sorption of toluene vapor and sorption followed by swelling should be the possible contributions when SU-8 polymer is exposed to toluene vapor.

To fully rule out any effect of condensation on the SU-8 polymer, the QCM system was connected to a water chiller that was used to control the temperature of the QCRs (the temperature was set to 12°C). The temperature-controlled QCRs were then exposed to toluene vapor to observe the condensation of toluene gas molecules onto the surface of the QCRs, as shown in Figure S23b. Here, the experiment was conducted in a glove box with low humidity (relative humidity of 20%) to avoid the condensation of the water vapor. In Figure S23b, the QCR sample with the SU-8 polymer layer shows a response to toluene vapor at the initial stage (room

temperature). After turning on the cooling system, the estimated thickness of the QCR with the SU-8 polymer layer still increases at a constant rate and saturates to approximately 10 Å in 350 s, and then a sudden increase occurs at about 630 s. Because this sudden increase was not observed for the QCR with the SU-8 polymer layer in the absence of toluene vapor at a low temperature (see inset of Figure S23b), we attribute this response to the condensation of the toluene gas molecules (we could also confirm condensation of toluene on the bare QCR). It is noted that the low increase rate at the initial stage should correspond to the sorption of toluene vapor or sorption followed by swelling of the SU-8 polymer. According to these results, we could confirm that the sensing mechanism of the proposed BSW lasers should be related to the effect of sorption or sorption followed by swelling of the SU-8 polymer ridge.

Figure S23. The change in the thicknesses of QCRs without (bare QCR in red) and with a SU-8 polymer coating (SU-8 polymer coated QCR in blue) in response to toluene vapor. (a) Responses to toluene and recoveries at room temperature. (b) Response to toluene when the temperature of the QCR is controlled (note the slow increase rate attributed to sorption of toluene vapor or sorption followed by swelling and the fast rate attributed to condensation). Inset: the full response of QCRs with a SU-8 polymer coating in the absence (red) and presence (blue) of toluene vapor.

According to the QCM experiment, we concluded that both the sorption of toluene vapor and sorption followed by swelling of the SU-8 polymer ridge should be possible sensing mechanisms for the proposed BSW lasers. Thus, the theoretical investigation of the environmental sensitivity of the proposed BSW lasers should include both the sorption and the swelling effect of the SU-8 polymer ridge on the lasing behavior in the presence of toluene vapor. Accordingly, we have added the simulated wavelength shifts of the investigated BSW ring laser with an additionally deposited toluene layer ($n = 1.50$, considering only the sorption of toluene vapor) onto the surface of the SU-8 polymer ridge, which is obtained by using eigenfrequency analysis for an axisymmetric model, as displayed in Figure S24a. The simulated wavelength shifts of the photonic laser ($d = 520$ nm) with deposited toluene layer are also displayed (Figure S24b). Because the refractive index of the deposited toluene layer is close to that of the SU-8 polymer, both the sorption and swelling effect (considering the increase in the ridge size Δa , as displayed in Figure S24c) exhibit similar wavelength shifts for the BSW ring laser. The deposition of the toluene layer of 0.14 nm (Figure S24a) and the increase in the ridge size of 0.1 nm (Figure S24c) or a change in the refractive index of 0.00023 (Figure S24d) all correspond to a 0.05 nm wavelength shift, which is of the same order as the observed wavelength shift in Figure 6b. Regarding the investigation of the swelling effect, we found that the simulated values for the changes that can explain the lasing shifts are of the same order than those measured for the swelling of non-baked SU-8 polymer³. It is also worth mentioning that the increase in the ridge thickness contributes 97% of the wavelength shift, whereas the change in the ridge width contributes only 3%, thus indicating a high sensitivity of the BSW mode due to its evanescent nature. According to these investigations, we conclude that, although it is difficult to distinguish between the sensing mechanisms including the sorption of toluene vapor and the sorption

followed by the swelling effect of the SU-8 polymer, a high sensitivity for the BSW lasers could still be obtained.

Figure S24. The simulated wavelength shifts of (a) the BSW ring lasers and (b) the photonic ring lasers ($d = 520$ nm) with additionally deposited toluene layer ($n = 1.50$, considering only the sorption of toluene vapor) onto the surface of the SU-8 polymer ridge. (c, d) The simulated wavelength shifts of the BSW ring lasers with respect to (c) the increase in ridge size Δa and (d) the increase in refractive index Δn (assuming the absorption of toluene vapor without a change in size), simulated separately. (e) The simulated wavelength shifts of the photonic ring lasers ($d = 520$ nm) with respect to the increase in ridge size Δa . The ring diameter for both BSW laser and photonic laser is $100 \mu\text{m}$. The BSW platform comprises five $\text{TiO}_2/\text{SiO}_2$ pairs having respective thicknesses of $82 \text{ nm}/170 \text{ nm}$.

The revised part of the manuscript is provided below:

(Line 1 page 19) “*Finally, we experimentally performed a sensitivity investigation to environmental changes of the proposed BSW lasers by monitoring the lasing wavelength in the presence of toluene vapor (experimental procedure described in the Supporting Information).*”

(Line 9 page 19) “*The lasing peak shift is explained by the sorption of toluene vapor only or the sorption followed by swelling of the SU-8 resist due to toluene molecules diffusing into the SU-8 polymer⁵⁸ (regarding the clarification of the sensing mechanism, see Figure S23). The sorption of toluene vapor onto the surface of SU-8 polymer increases the refractive index of the adjacent medium; the swelling of SU-8 polymer is known to increase the resist volume and refractive index, thus shifting the lasing peaks⁵⁹.*”

(Line 17 page 19) “*A total redshift of 0.05 nm is finally observed which can correspond to a sorbed toluene layer of 0.14 nm, an increase in the ridge size of 0.1 nm or a change in the refractive index of 0.00023 (see Figure S24), thus implying the detection of a very small perturbation of the BSW laser structure and suggesting possible applications of the BSW lasers as ultra-sensitive devices.*”

Reference

1. Wouters, K., Puers, R. Diffusing and swelling in SU-8: insight in material properties and processing. *J. Micromech. Microeng.* **20**, 095013 (2010).
2. Rianjanu, A. *et al.* Swelling behavior in solvent vapor sensing based on quartz crystal microbalance (QCM) coated polyacrylonitrile (PAN) nanofiber. *IOP Conf. Ser.: Mater. Sci. Eng.* **367**, 012020 (2018).
3. Saunders, J. E. *et al.* Quantitative diffusion and swelling kinetic measurements using large-angle interferometric refractometry. *Soft Matter* **11**, 8746–8757 (2015).

2. The numerical analysis of the losses in BSW resonators does not look self-consistent. First, as indicated in the main text, Q -factors are calculated using eq. (2) and (3) for BSW ring cavities, implying that Q_{leak} must be exactly equal to $Q_{\text{leak-surr}}$. If the values in eq. (3) are calculated for straight waveguides, this should be clearly reflected in the main text of the manuscript. However, in this case, the statement «the light leakage to the surrounding is indeed a dominant loss» is incorrect. As an example, for $D = 100$ μm and waveguide width of 2.5 μm , $Q_{\text{leak}} = 20000$ (Fig. 5b) and $Q_{\text{leak-surr}} = 40000$ (Fig. S14c). Thus, $Q_{\text{leak-curv}} = 40000$, i.e., for the largest cavity, the losses due to curvature is equal to losses due to leakage in surrounding and will only increase with decreasing diameter. However, in the case of $D = 100$ μm and waveguide width of 1.5 μm , $Q_{\text{leak}} \approx Q_{\text{leak-surr}} \approx 15000$ meaning no losses due to curvature. The authors should discuss this discrepancy.

We have added a discussion section “*Detailed discussions about the Q_{leak} of the BSW ring cavities*” and Figures S16 and S17 in the Supporting Information and revised the manuscript accordingly. We have also revised the manuscript to state that we evaluated the light leakage to the surrounding via Equation 3 by considering a straight waveguide which sustains the propagating BSW mode.

We indeed evaluated the quality factor related to the leakage of light to the air or the substrate ($Q_{\text{leak-surr}}$ is calculated by using Equation 3) by considering a straight waveguide, which sustains the propagating BSW mode. However, as noted by the reviewer, the values of $Q_{\text{leak-curv}}$ of the BSW ring cavities having the same diameter of $D = 100$ μm but with the different ridge widths of $w = 1.5$ and 2.5 μm seem different. To clarify this discrepancy, we simulated the Q_{leak} of the BSW ring cavities for the ridge with non-vertical sidewalls as a function of the width for different diameters,

using eigenfrequency analysis for an axisymmetric model, and compared the results with the $Q_{\text{leak-surr}}$, as displayed in Figures S16a–d. The Q_{leak} values are found to be independent of the ridge width for the BSW ring cavities having large widths. Once the ridge width decreases such that the value of $Q_{\text{leak-surr}}$ is reduced to be comparable to that of the Q_{leak} , Q_{leak} starts to decrease with a similar trend than that of the decrease in the $Q_{\text{leak-surr}}$ for small ridge widths. This behavior can be understood from the simulated electric field norm distributions for the WGM resonances of the BSW ring cavities having various diameters and ridge widths, as shown in Figures S16e–h. For the case that the ridge width is large enough (i.e., $w = 2.5 \mu\text{m}$), the WGM modes in the BSW ring cavities are readily supported by the ring cavity. Note that these distributions of the WGM modes are pushed outward in the radial direction, and the distribution becomes smaller with decreasing diameter due to the effect of the curvature. When the ridge width further decreases from $w = 2.5 \mu\text{m}$, the WGM modes are still supported by the ring cavity and the distributions remain the same. In this case, the Q_{leak} is independent of the ridge width, suggesting that the Q_{leak} is evaluated by only considering the WGM mode distribution in the eigenfrequency analysis. When the ridge width is decreased until the inward ridge edge becomes close to the WGM mode distribution, then the $Q_{\text{leak-surr}}$ becomes comparable to the Q_{leak} . A further decrease in the ridge width results in a mode distribution with an electric field distribution similar to that of a guided BSW mode within a ridge waveguide (see Figures S16i and S16j), and the Q_{leak} is found to be similar to the $Q_{\text{leak-surr}}$ for small ridge widths. According to these results, we conclude that the simulated Q_{leak} using eigenfrequency analysis for an axisymmetric model corresponds to the $Q_{\text{leak-surr}}$ only when the mode distributions are the same, that is, the WGM mode distributions as obtained by eigenfrequency analysis become similar to the distributions for the mode analysis.

Figure S16. The detailed investigation of the Q_{leak} of the BSW ring cavities with various diameters and ridge widths. The BSW platform comprises five $\text{TiO}_2/\text{SiO}_2$ pairs having respective thicknesses of 82 nm/170 nm. (a–d) The comparisons between the simulated $Q_{\text{leak-surr}}$ of the BSW modes using mode analysis (diameter independent results) and the simulated Q_{leak} of the ring laser cavities having diameters of (a) $D = 100 \mu\text{m}$, (b) $D = 75 \mu\text{m}$, (c) $D = 50 \mu\text{m}$, and (d) $D = 30 \mu\text{m}$ obtained by eigenfrequency analysis for an axisymmetric model. (e–h) The calculated electric field norm distributions for the BSW ring laser cavities with (e) $D = 100 \mu\text{m}$, (f) $D = 75 \mu\text{m}$, (g) $D = 50 \mu\text{m}$, and (h) $D = 30 \mu\text{m}$ obtained by eigenfrequency analysis for an axisymmetric

model. Left: the distributions for the ridge width $w = 2.5 \mu\text{m}$; middle: the distributions for ridge width at which the $Q_{\text{leak-surr}}$ is approximately equal to Q_{leak} ; right: the distributions for the ridge width $w = 1 \mu\text{m}$. (i) The calculated electric field norm distribution of the BSW mode for $d = 136 \text{ nm}$ and $w = 1 \mu\text{m}$ using mode analysis. (j) The calculated electric field norm distributions of the ring laser cavities ($d = 136 \text{ nm}$, $w = 1 \mu\text{m}$) for $D = 100 \mu\text{m}$ (left) and $D = 30 \mu\text{m}$ (right), respectively, using eigenfrequency analysis for an axisymmetric model.

From the above investigation, it is concluded that Q_{leak} and $Q_{\text{leak-surr}}$ give similar results when the BSW ring cavities possess the same distribution as simulated in an axisymmetric model (eigenfrequency analysis) and straight waveguide model (mode analysis). Thus, we assumed that the lateral radiation leakage due to curvature should be almost zero ($1/Q_{\text{leak-curv}} \approx 0$). To verify this assumption, a simulation model combined with effective index method (EIM) and two-dimensional (2D) eigenfrequency analysis (COMSOL software) is employed to simulate the $Q_{\text{leak-curv}}^{-1}$, as displayed in Figures S17a–c. In this model, a 2D structure of a BSW ring cavity is divided into two regions, including the BSW platform without (bare, blue) and with (ridge, red) a 136-nm-thick top SU-8 polymer ridge (Figure S17a). The effective indices of these two regions are calculated using the mode analysis, as displayed in Figure S17b. Note that the lateral radiation leakage due to the effect of curvature is only related to the effective refractive index ($n_{\text{effective}}$) difference between such two regions. It is also noted that the effect of the oblique sidewall has been taken into account in the mode analysis (regarding the Q_{leak} investigation for the SU-8 polymer ridge with vertical sidewalls, see Figures S17d–f). Figure S17c shows the simulated $Q_{\text{leak-curv}}$ of the BSW ring cavities ($d = 136 \text{ nm}$, $w = 1.5$ and $2.5 \mu\text{m}$) for different diameters. The $Q_{\text{leak-curv}}$ values of the BSW ring cavities are relatively high ($Q_{\text{leak-curv}}$

$\geq 10^{11}$), suggesting that the lateral radiation leakage due to curvature can be neglected. Based on these investigations, the light leakage to the surrounding is indeed a dominant loss for our proposed BSW ring cavities.

Figure S17. Investigation into the $Q_{\text{leak-curr}}$ of the BSW laser ring cavities having various diameters. (a, b) Schematic representations of the 2D eigenfrequency analysis for simulating the $Q_{\text{leak-curr}}$: (a) top view and (b) cross-sectional view of the simulation model for calculating the effective indices for bare (blue in figure a) and ridge (red in figure a) regions. The BSW platform comprises five $\text{TiO}_2/\text{SiO}_2$ pairs having respective thicknesses of 82 nm/170 nm. (c) The simulated $Q_{\text{leak-curr}}$ of the BSW ring laser cavities ($d = 136 \text{ nm}$, $w = 1.5$ and $2.5 \mu\text{m}$) having various diameters. (d, e) The comparisons between the simulated $Q_{\text{leak-surr}}$ of the BSW modes using mode analysis (diameter

independent results) and the simulated Q_{leak} of the ring laser cavities having diameters of (d) $D = 100 \mu\text{m}$ and (e) $D = 30 \mu\text{m}$ obtained by eigenfrequency analysis for an axisymmetric model. Here, the sidewall of the SU-8 polymer ridge width is vertical. (f) The simulated $Q_{\text{leak-curv}}$ of the BSW ring laser cavities ($d = 136 \text{ nm}$, $w = 1.5$ and $2.5 \mu\text{m}$ with vertical sidewall) having various diameters.

The revised part of the manuscript is provided below:

(Line 1 page 16) *“Also, the Q_{leak} values of the BSW ring cavities increased with increasing diameter, due to the reduced curvature effect when the light propagates along a curved ridge with smaller curvature for the WGM resonances.”*

(Line 1 page 17) *“To investigate the light leakage to the surrounding, $Q_{\text{leak-surr}}$ is evaluated using the propagation losses of the corresponding guided BSW mode in a straight ridge waveguide²¹, and is defined and expressed via the mode effective indices ($n_{\text{effective}} + ik_{\text{effective}}$) by⁵⁶...”*

(Line 10 page 17) *“The detailed investigation reveals that the light leakage to the surrounding is indeed a dominant loss (see Figures S16 and S17) so that their Q_{leak} exhibit a weak ridge width dependence, which is different from the ridge width dependence on the lasing thresholds of the BSW lasers (Figure 5b).”* (Please note that the explanation for the lasing threshold dependency with the width is provided just after the above sentence in the manuscript: scattering via surface roughness is found to explain the lasing threshold.)

Reference

1. Menotti, M., Liscidini, M. Optical resonators based on Bloch surface waves. *J. Opt. Soc. Am. B* **32**, 431–438 (2015).

3. Technical comments.

3.1. The correctness of the citation should be carefully checked:

3.1.1. The WGM size dependence of the lasing wavelength is proved by Ref. 51, in which a very specific system of spherical microdroplet-lasers is studied. Could the authors refer to works where the size dependence effect was studied in WGM lasers on photonic platforms and briefly explain this effect in the manuscript?

We have added reference 52 to our manuscript related to the size dependence of the lasing spectrum and added some descriptions of the cited works as follows.

(Line 15 page 12) *“This size dependence is attributed to two possible reasons. First, the lasing wavelength range can vary due to the change in the wavelength for the maximum out-coupled lasing light intensity from the ring cavity, which shifts toward longer wavelengths when the ring diameter is increased⁵¹. Second, the variation in the lasing wavelength range can be explained by the intrinsic self-absorption in the ring cavity⁵².”*

Indeed the size dependence of the lasing spectrum was studied for spherical microdroplet-lasers in reference 51, which is different from our proposed BSW ring lasers. However, from the mode spacing analysis of the lasing modes, the WGM resonance mechanism for these two systems should be similar. Thus, we believe that it should be reasonable to attribute the size dependence effect to the change in the wavelength for the maximum out-coupled lasing light intensity from the ring cavity, as reported in reference 51.

Besides, a past study has also reported the size dependence of the lasing spectrum range from on-chip WGM lasers¹, and this effect is explained by the self-

absorption when light propagates along the cavity dimension. We suppose that such self-absorption provides another possible mechanism for the size dependence effect. Therefore, we have added this study as reference 52 to our manuscript.

Reference

1. Liu, X. *et al.* Periodic organic–inorganic halide perovskite microplatelet arrays on silicon substrates for room-temperature lasing. *Adv. Sci.* **3**, 1600137 (2016).

3.1.2. Incorrect reference 53 before the eq. (1).

We have revised the incorrect reference to our manuscript related to the total quality factor of an optical cavity in terms of the different loss mechanisms which can be expressed by Equation 1 (Reference 54: Krämmer, S. *et al.* Size-optimized polymeric whispering gallery mode lasers with enhanced sensing performance. *Opt. Express* **25**, 7884–7894 (2017)).

3.1.3. There are no data on the pump fluence and the pump beam size in Ref. S21 in supporting information.

We have added a footnote in **Table S2** to indicate how we estimated the lasing threshold for the on-chip WGM laser of reference S21.

Although the pump beam size has not been specified in reference S21 discussed in our Supporting Information, the threshold pump energy (24 nJ per pulse) has actually been given from the input-output curve of an optically pumped dye-doped microdisk laser. We supposed that the pumped area should be slightly larger than the size of the WGM cavity. Thus, the pump beam size is assumed to be approximately

50 μm . According to the information mentioned above, we calculated the pump energy density to be approximately 12 $\mu\text{J}/\text{mm}^2$.

Table S2. On-chip WGM lasers based on dye-doped polymers.

Gain medium (method)	Waveguide mode	Guiding structure thickness	Diameter	Threshold	Pulse duration of pump laser	Year	Ref.
BEH-PPV	Photonic mode	NA	100 μm	10 $\mu\text{J}/\text{mm}^2$	10 ns	1998	[18]
R6G doped SU-8	Photonic mode	48 μm	(trapezoid) round-trip length 292 μm	25 $\mu\text{J}/\text{mm}^2$	5 ns	2005	[19]
R6G doped PMMA	Photonic mode	1 μm	40 μm	2.5 $\mu\text{J}/\text{mm}^2$	15 ns	2010	[20]
PM597 doped resist	Photonic mode	1 μm	47 μm	12 $\mu\text{J}/\text{mm}^2$ *	5 ns	2011	[21]
PM597 doped PMMA	Photonic mode	~ 1 μm	50 μm	0.3 $\mu\text{J}/\text{mm}^2$	20 ns	2015	[22]
R6G doped SU-8	Photonic mode	12 μm	300 μm	0.2 $\mu\text{J}/\text{mm}^2$	10 ns	2015	[23]
R6G-doped TZ-001	Photonic mode	30 μm	220 μm	9.3 $\mu\text{J}/\text{mm}^2$	5 ns	2017	[24]
R6G doped SU-8	Photonic mode	30 μm	220 μm	2.5 $\mu\text{J}/\text{mm}^2$	5 ns	2018	[25]
R6G doped SU-8	Photonic mode	0.24 μm (at the cutoff of the photonic mode)	100 μm	100 $\mu\text{J}/\text{mm}^2$	0.5 ns	2023	This work
R6G doped SU-8	BSW mode	0.14 μm	100 μm	6.7 $\mu\text{J}/\text{mm}^2$	0.5 ns	2023	This work

*The pumped spot size is assumed to be 50 μm .

3.2. As the WGM lasing threshold strongly depends on the WGM diameter, the latter should be added in the Table S2.

We thank the Reviewer for this suggestion. We have added the diameter of the WGM lasers in Table S2.

Table S2. On-chip WGM lasers based on dye-doped polymers.

Gain medium (method)	Waveguide mode	Guiding structure thickness	Diameter	Threshold	Pulse duration of pump laser	Year	Ref.
BEH-PPV	Photonic mode	NA	100 μm	10 $\mu\text{J}/\text{mm}^2$	10 ns	1998	[18]
R6G doped SU-8	Photonic mode	48 μm	(trapezoid) round-trip length 292 μm	25 $\mu\text{J}/\text{mm}^2$	5 ns	2005	[19]
R6G doped PMMA	Photonic mode	1 μm	40 μm	2.5 $\mu\text{J}/\text{mm}^2$	15 ns	2010	[20]
PM597 doped resist	Photonic mode	1 μm	47 μm	12 $\mu\text{J}/\text{mm}^2$ *	5 ns	2011	[21]
PM597 doped PMMA	Photonic mode	~ 1 μm	50 μm	0.3 $\mu\text{J}/\text{mm}^2$	20 ns	2015	[22]
R6G doped SU-8	Photonic mode	12 μm	300 μm	0.2 $\mu\text{J}/\text{mm}^2$	10 ns	2015	[23]
R6G-doped TZ-001	Photonic mode	30 μm	220 μm	9.3 $\mu\text{J}/\text{mm}^2$	5 ns	2017	[24]
R6G doped SU-8	Photonic mode	30 μm	220 μm	2.5 $\mu\text{J}/\text{mm}^2$	5 ns	2018	[25]
R6G doped SU-8	Photonic mode	0.24 μm (at the cutoff of the photonic mode)	100 μm	100 $\mu\text{J}/\text{mm}^2$	0.5 ns	2023	This work
R6G doped SU-8	BSW mode	0.14 μm	100 μm	6.7 $\mu\text{J}/\text{mm}^2$	0.5 ns	2023	This work

*The pumped spot size is assumed to be 50 μm .

3.3. As it turned out that experimental values of PC layer thicknesses differ from that used in initial calculations, it should be specified for which PC parameters the calculations were carried out in each case (especially for Fig. 5).

Thank you for your suggestion. We have specified the thicknesses of TiO_2 and SiO_2 layers for each simulation in the figure captions of **Figures 1 and 5** in the manuscript and **Figures S3, S6, S11, S14, S15, and S24** in the Supporting Information.

3.4. For what length of waveguides is the coupling efficiency calculated in Fig. S19? I do not believe that the coupling efficiency was underestimated. On the one hand, the coupling can be enhanced due to the shift of the waveguide mode toward the outer side of the curved waveguide. On the other hand, as the curved waveguide moves away from the straight one, the coupling efficiency decreases. The more rigorous calculations are needed.

We have revised the simulation for the coupling between a BSW ring laser cavity and a BSW waveguide. The manuscript text in the section reporting the “*Coupling between a BSW ring laser cavity and a waveguide*” and **Figure S21** in the Supporting Information have been revised accordingly.

Thank you for pointing out this issue. We agree that the simplified simulation model used to estimate the coupling efficiency by considering two parallel straight ridge waveguides may be inappropriate. In this simplified model, the BSW mode couples to the adjacent waveguide through the gaps with a coupling length of 10 μm , while in the real case the efficient coupling only occurs near the vertex of the curved waveguide. Therefore, we have revised the simulation model to make a more rigorous estimation of the coupling efficiency. As displayed in Figures S21a and S21b, the simulation model is set to be composed of a curved waveguide (a fraction of the BSW ring laser cavity having a diameter of 30 μm) and a straight waveguide, and the smallest distance between them is defined as the coupling gap. A BSW mode is launched from one end of the input curved ridge waveguide and propagates along the tangential direction. Some portion of the guiding BSW mode will couple to the adjacent waveguide and then propagates to the end of the output ridge waveguide. Then, the coupling efficiency is estimated as the ratio between the power at the output ridge and the input ridge ($P_{\text{out}}/P_{\text{in}}$). Figure S21c shows the simulated electric field

norm distribution for the coupling between the SU-8 polymer ring cavity and the ridge waveguide, and the coupling efficiency for the BSW laser as a function of the gap is reported in Figure S21d. The coupling efficiency increases with decreasing the gap, and a value of approximately 0.2% is obtained for a gap of 0.050 μm . Note that these results were almost the same as those obtained by launching the BSW mode in the guiding slab and coupling to the SU-8 polymer ring cavity. In addition, the estimated coupling efficiencies are of the same order than that of a structure realizing standard evanescent coupling¹, thus suggesting the validity of our simulation model. Although the estimated coupling efficiency is relatively low, we believe that more efficient couplings between a BSW ring laser and a BSW waveguide can be obtained via improved coupling designs such as wrapped waveguides². Accordingly, the applicability of the proposed BSW lasers in integrated optical circuits is established so that future applications of BSW lasers on a chip such as optical modulators and logic devices can be realized.

Figure S21. Investigation into the coupling between a BSW ring laser cavity and a BSW guiding slab. (a, b) Schematic representation of the (a) coupling between a BSW ring laser cavity and a straight BSW waveguide, and (b) cross-sectional view of the simulation model. The width of the SU-8 polymer ridge is $1.5 \mu\text{m}$. (c, d) The simulated (c) electric field norm distribution for coupling between the BSW ring laser cavity and the waveguide and (d) the estimated coupling efficiency as a function of the gap.

Reference

1. Cegielski, P. J. *et al.* Monolithically integrated perovskite semiconductor lasers on silicon photonic chips by scalable top-down fabrication. *Nano Lett.* **18**, 6915–6923 (2018).
2. Yang, K. Y. *et al.* Bridging ultrahigh-Q devices and photonic circuits. *Nat. Photon.* **12**, 297–302 (2018).

3.5. According to Snell's law, the refraction angle θ inside TiO₂ should be approximately 29 degrees instead of 36.1 degrees.

Thank you for pointing out this mistake. We have revised the refraction angle θ of light within the TiO₂ layer to be 29° in the Supporting Information.

[1] Konopsky, V. N. & Alieva, E. V. Photonic crystal surface waves for optical biosensors. *Anal. Chem.* 79, 4729–4735 (2007).

[2] Sinibaldi, A. et al. Direct comparison of the performance of Bloch surface wave and surface plasmon polariton sensors. *Sensors Actuators, B Chem.* 174, 292–298 (2012).

[3] Descrovi, E., Coupling of surface waves in highly defined one-dimensional porous silicon photonic crystals for gas sensing applications, *Appl. Phys. Lett.* 91, 241109 (2007).

[4] Michelotti, F., et al. Fast optical vapour sensing by Bloch surface waves on porous silicon membranes, *Physical Chemistry Chemical Physics*, 12(2), 502-506 (2010).

(The reference list from Reviewer 4 was very helpful to revise our manuscript)

REVIEWERS' COMMENTS

Reviewer #4 (Remarks to the Author):

I am grateful to the authors for the careful consideration of all the issues raised and the great research done. Now all the statements have been proven and the manuscript deserves publication in Nature Communications.